# Short-term molecular consequences of chromosome mis-segregation for genome stability

Lorenza Garribba[1,8], Giuseppina De Feudis[1,8], Valentino Martis[1], Martina Galli[2], Marie Dumont[3], Yonatan Eliezer[4], René Wardenaar [5], Marica Rosaria Ippolito[1], Divya Ramalingam Iyer [6], Andréa E. Tijhuis[5], Diana C. J. Spierings [5], Michael Schubert [5], Silvia Taglietti[1], Chiara Soriani [1], Simon Gemble[3], Renata Basto [3], Nick Rhind [6], Floris Foijer [5], Uri Ben-David [4], Daniele Fachinetti [3], Ylli Doksani [2] & Stefano Santaguida [1,7] ✉

Chromosome instability (CIN) is the most common form of genome instability and is a hallmark of cancer. CIN invariably leads to aneuploidy, a state of karyotype imbalance. Here, we show that aneuploidy can also trigger CIN. We found that aneuploid cells experience DNA replication stress in their first S-phase and precipitate in a state of continuous CIN. This generates a repertoire of genetically diverse cells with structural chromosomal abnormalities that can either continue proliferating or stop dividing. Cycling aneuploid cells display lower karyotype complexity compared to the arrested ones and increased expression of DNA repair signatures. Interestingly, the same signatures are upregulated in highly-proliferative cancer cells, which might enable them to proliferate despite the disadvantage conferred by aneuploidy-induced CIN. Altogether, our study reveals the short-term origins of CIN following aneuploidy and indicates the aneuploid state of cancer cells as a point mutation-independent source of genome instability, providing an explanation for aneuploidy occurrence in tumors.

Chromosomal instability (CIN), a condition of continuous chromosome mis-segregation, is a pervasive feature of tumors[1,2]. CIN confers enhanced evolutionary capabilities on cancer cells by increasing intratumor heterogeneity and by enabling chemoresistance[3–6]. CIN leads invariably to aneuploidy, a state of karyotype imbalances, found in more than 90% of solid tumors and about 65% of blood cancers[1]. The presence of aneuploid karyotypes leads to several detrimental defects, including proteotoxic stress[7–9], metabolic alterations[10] and induction of DNA damage[7,11–14].

Importantly, karyotype aberrations strongly correlate with poor patient prognosis[2]. This might be due to the fact that specific aneuploidies could confer a proliferative advantage, thus fueling tumorigenesis[5,6] and promoting survival under sub-optimal conditions[3,4]. Such an advantage could be explained by the possibility that aneuploidy induces CIN (and, more broadly, genome instability), which might enable a continuous sculpting of the genome, eventually leading to cumulative haploinsufficiency and triplosensitivity[15,16] of genes crucial for sustained proliferation. In

[1]Department of Experimental Oncology at IEO, European Institute of Oncology IRCCS, Via Adamello 16, 20139 Milan, Italy. [2]IFOM ETS - The AIRC Institute of Molecular Oncology, via Adamello 16, 20139 Milan, Italy. [3]Institut Curie, PSL Research University, CNRS, UMR144 Paris, France. [4]Department of Human Molecular Genetics and Biochemistry, Faculty of Medicine, Tel Aviv University, Tel Aviv, Israel. [5]European Research Institute for the Biology of Ageing, University of Groningen, University Medical Center Groningen, 9713 AV Groningen, the Netherlands. [6]Department of Biochemistry and Molecular Biotechnology, University of Massachusetts Chan Medical School, 364 Plantation Street, Worcester, MA 01605, USA. [7]Department of Oncology and Hemato-Oncology, University of Milan, Via Santa Sofia 9/1, 20122 Milan, Italy. [8]These authors contributed equally: Lorenza Garribba, Giuseppina De Feudis. ✉e-mail: stefano.santaguida@ieo.it

agreement with this idea, studies in yeast have demonstrated that gain of a single chromosome leads to defective DNA damage repair[12]. Further, aneuploid strains often divide in presence of unrepaired DNA, which triggers chromosomal translocations[17]. Similar observations were made in higher eukaryotes[18]. For example, a comparison between trisomic and diploid human cells has revealed that aneuploid cells are characterized by increased frequency of lagging chromosomes in anaphase. Thus, this evidence points at aneuploidy as an instigator of genome instability[16]. It is plausible that this instability is due to the strong impact of karyotype abnormalities on gene expression and protein homeostasis. In fact, aneuploid cells were found to display imbalances in factors critical for DNA replication (such as MCM2-7), DNA repair and mitosis[13], processes that are all fundamental for the maintenance of genome integrity. In line with this possibility, previous studies have revealed that aneuploid cells exhibit an increased S-phase duration, display reduced DNA replication fork rate and increased fork stalling[13,14]. Due to the intrinsic genomic instability and other stresses typically associated with aneuploidy, cells with abnormal karyotypes often exhibit delayed cell cycle progression. In some cases, they even lose their proliferative capacity and stop dividing[1,14,19], resulting in their reduced sensitivity to chemotherapies[20,21].

Given the high prevalence of unbalanced karyotypes in tumors and its impact on the proliferation of cancer cells[3–6], elucidating the contribution of aneuploidy to genome instability, deciphering the molecular mechanisms by which it occurs and deconvolving its cellular consequences remain of paramount importance in cancer biology.

Here, by inducing controlled chromosome mis-segregation in otherwise pseudo-diploid human cells, we set out to identify the origins of genome instability in aneuploid cells and to understand whether protective mechanisms operate to preserve genome integrity. Our data indicate that in the first S-phase following chromosome mis-segregation, aneuploid cells fire dormant replication origins through a DDK-dependent mechanism and complete replication of genomic loci through mitotic DNA synthesis (MiDAS). Importantly, those pathways, acting both in interphase and mitosis, are crucial for aneuploid cells to protect them against further genome instability, thus maintaining low levels of CIN. We also show that the DNA damage associated with aneuploidy can be distributed asymmetrically between daughter cells during cell division and this, at least partially, can explain why some cells (i.e. those who have inherited most of the damage) stop dividing. By establishing a method for the separation of arrested and cycling aneuploid cells, we found that cycling aneuploid cells exhibit increased expression of DNA repair genes. Interestingly, the same transcriptional signature was upregulated in cancer cells characterized by high proliferative capacity. We speculate that elevated expression of DNA damage repair genes in highly proliferative cancers is able to help them counteracting the burden associated with genome instability, allowing them to benefit from a continuous reshuffling of the karyotype, which is crucial to sustain enhanced proliferation[3,4]. Finally, we speculate that DDK-mediated origin firing and MiDAS are crucial for limiting DNA damage, and interfering with those pathways might provide novel therapeutic interventions in cancer therapy. An example of this is given by ongoing clinical trials involving agents inhibiting DDK-mediated origin firing (e.g., ClinicalTrials.gov Identifier: NCT03096054 and NCT05028218). Thus, our work might help in the stratification of patients who could benefit from those therapeutic approaches, indicating that those treatments might be particularly effective in tumors with high proliferative capacity.

## Results

### Identification of mechanisms responsible for tolerance to aneuploidy-induced replication stress

Aneuploidy is associated with increasing genome instability[16], affecting the fidelity of both genome replication and segregation. To dissect the

mechanisms through which aneuploid cells seek to limit this instability and thus keep proliferating, we quantified the direct effects of aneuploidy on genome integrity. For this, we analyzed chromosome aberrations immediately after the induction of mitotic errors (1st mitosis) and after one cell cycle (2nd mitosis). To this aim, we synchronized untransformed and genomically-stable, pseudo-diploid hTERT RPE-1 cells with thymidine at the G1/S border (Supplementary Fig. 1a, b) and, after release into the cell cycle, pulsed them with DMSO (vehicle control) or reversine, an Mps1 inhibitor widely-used to generate aneuploid cells as a consequence of chromosome segregation errors[22]. Cells were then either harvested for karyotype analysis of the 1st mitosis or, after DMSO or reversine wash-out, allowed to continue in the cell cycle, and then harvested for the same purpose in the 2nd mitosis (Fig. 1a). To rule out the possibility that the synchronization method and/or reversine treatment would impact on DNA damage thus affecting the results of our experiments, we evaluated the consequences of thymidine block and release in presence of reversine using γH2AX as a marker of DNA damage (Supplementary Fig. 1c–e). We found that, although thymidine block induced mild levels of DNA damage, this was fully repaired 12 h after washout with a similar kinetics in cells exposed to Mps1 inhibitor or vehicle control (Supplementary Fig. 1c–e). We also treated cells with doxorubicin to induce DNA damage and observed no statistically significant differences in the DNA repair capacity of cells treated with the Mps1 inhibitor or vehicle control (Supplementary Fig. 1c, f, g). This shows that treatment with reversine does not affect cell capacity to repair the damage. Thus, by using this approach, analysis of 1st mitosis provided a measurement of the degree of chromosome aberrations directly caused by aneuploidy induction, whereas quantification of 2nd mitosis allowed for the estimation of genome alteration as a consequence of harboring aneuploid karyotypes. By using multi-color FISH (mFISH), we found abnormal events – including gains, losses and translocations – in both the 1st and 2nd mitoses (Fig. 1b, c and Supplementary Fig. 1h, i). Importantly, the percentage of cells harboring more than 10 abnormal events more than tripled from the 1st to the 2nd mitosis (Fig. 1b, c) indicating that the aneuploid state per se negatively impacts genome stability. However, we also note that Mps1i pulse in the 1st mitosis could affect chromosome stability in a long-term and aneuploidy-independent manner, through not-yet-identified mechanisms unrelated to chromosome mis-segregation.

Notably, numerical aneuploidies accounted for most of the measured aneuploidy events, both in the 1st and in the 2nd mitosis (Fig. 1d). Next, to decipher how aneuploidy affects genome integrity, we examined at high resolution the 1st S-phase of newly-generated aneuploid cells. For this, we synchronized cells at the G1/S border, released them in the presence of reversine or vehicle control, blocked them in late G1 with mimosine and released them in S-phase (Fig. 1e and Supplementary Fig. 2a, b). Similar to thymidine, mimosine leads to accumulation of mild levels of DNA damage that is repaired over time no matter whether cells had received reversine or the vehicle control in the previous 24 h (Supplementary Fig. 2c–e). For the study of the 1st S-phase of aneuploid cells obtained as above, we used three complementary approaches: (1) ultra-structural visualization of replication forks through electron microscopy (EM), (2) single-cell analysis of replication stress and DNA damage markers by immunofluorescence and (3) assessment of replication dynamics by DNA combing (Fig. 1e). These efforts led to three key observations. First, EM analysis of replication intermediates revealed an increase in reversed replication forks in aneuploid cells, compared to pseudo-diploid counterparts (Fig. 1f, g). Accumulation of these intermediates is associated with an increased frequency of replication fork stalling[23] and is consistent with previous observations of ongoing replication stress in aneuploid cells[13,14]. Further, aneuploid cells displayed increased levels of DNA replication stress and DNA damage markers such as FANCD2 (mean foci in control: 14.2 ± 1.9; aneuploid:

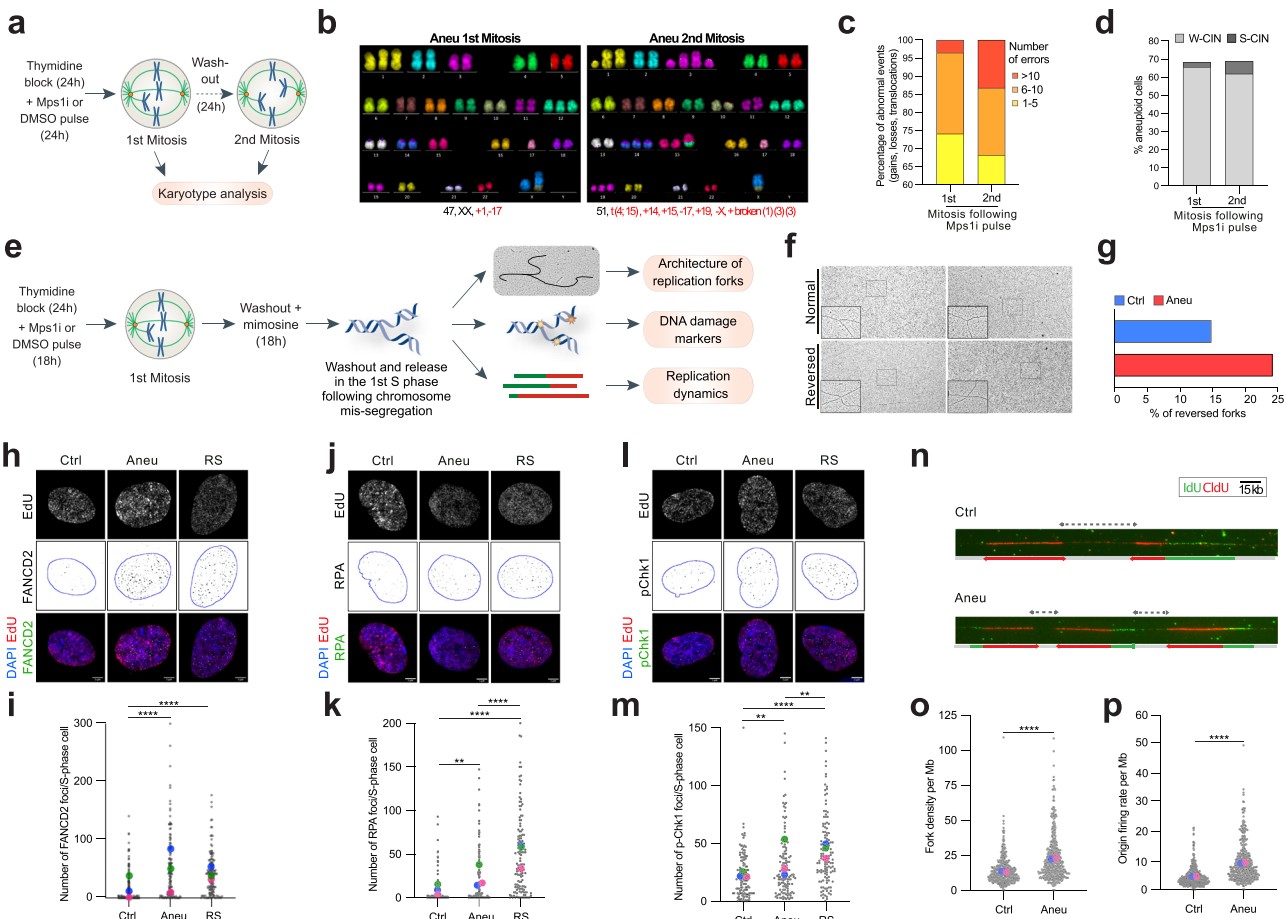

**Fig. 1 | Aneuploid cells accumulate increasing genome instability and display higher levels of DNA replication stress markers in S-phase. a** Experimental setup for the analysis of genome instability of cells obtained from the 1st and the 2nd mitosis. Karyotype aberrations were assessed by mFISH analysis (see "Methods" for more details). Representative mFISH images (**b**) of karyotypes obtained from the 1st (*n* = 68) and the 2nd mitosis (*n* = 98) in aneuploid cells and relative quantification (**c**). Y axis shown from 60 to 100% for clarity (from 0 to 60%−and above−cells have 1–5 abnormal events, as indicated in key legend). T(10,X) and +12 were excluded from the analysis as they are clonal in hTERT RPE-1 cells. **d** Quantification of aneuploid cells with either numerical (W-CIN) or structural (S-CIN) aneuploidy in aneuploid cells obtained from the 1st (*n* = 68) and the 2nd mitosis (*n* = 98).
**e** Schematic representation of the experimental approaches used for the study of the 1st S-phase after induction of chromosome missegregation. A short EdU pulse was performed before cell harvest in order to label S-phase cells to be analyzed by immunofluorescence. Representative images of normal and reversed replication forks (**f**) analyzed by electron microscopy and quantification (**g**) of the reversed ones in control (*n* = 108) and aneuploid (*n* = 95). Representative images (**h**) and quantification (**i**) of FANCD2 foci per S-phase cell in control (*n* = 181) and aneuploid (*n* = 148). **** indicates *p* < 0.0001. Representative images (**j**) and quantification (**k**) of RPA foci per S-phase cell in control (*n* = 133) and aneuploid (*n* = 136) cells.

** indicates *p* = 0.0015 and **** indicates *p* < 0.0001. Representative images (**l**) and quantification (**m**) of pChk1 foci per S-phase cell in control (*n* = 129) and aneuploid (*n* = 134) cells. ** indicates *p* = 0.0031 (Ctrl vs Aneu) or *p* = 0.0013 (Aneu vs RS) and **** indicates *p* < 0.0001. Cells treated with aphidicolin (RS, replication stress) were used as a positive control (*n* = 144 for (**i**), *n* = 137 for (**k**) and *n* = 131 for (**m**)).
**n** Representative images of DNA fiber analysis in control and aneuploid cells. o,p, Quantification of fork density per Mb (**o**) and origin firing rate (**p**) per Mb in control (*n* = 411) and aneuploid (*n* = 425) cells. **** indicates *p* < 0.0001. Ctrl, control (DMSO pulsed). Aneu, aneuploid cells (Mps1 inhibitor pulsed). RS, replication stress (aphidicolin treated cells). W-CIN, numerical chromosomal instability. S-CIN, structural chromosomal instability. Scale bars, 5 μm. LUT was inverted for FANCD2, RPA and pChk1 images. Blue borders in images are based on DAPI staining and define nuclei. Data are means of three biological replicates, except for the EM (one replicate) and the DNA fiber analysis (two replicates). Two-sided Chi square test was performed for data in (**c**) Two-tailed unpaired Student's *t* test was performed for data in (**i**, **k**, **m**, **o**, **p**). In graphs, average values for each biological replicate are shown by colored dots (each color corresponds to a different biological replicate). Source Data are provided as a Source Data file. Drawings of schemes were made by partially utilizing extracts of figures published elsewhere[1].

47.7 ± 4.6), RPA (mean foci in control: 8 ± 1.4; aneuploid 19.5 ± 2.7) and pChk1 (mean foci in control: 23 ± 1.8; aneuploid 33.3 ± 3.1) (Fig. 1h−m). Among them, the number of FANCD2 foci per S-phase cell was found to be even higher in aneuploid cells than in cells treated with the DNA replication inhibitor aphidicolin, used as a positive control (FANCD2 mean foci in aphidicolin-treated cells: 40.7 ± 3; RPA mean foci 49 ± 3.9; pChk1 mean foci 44.1 ± 2.6) (Fig. 1i, k, m and Supplementary Fig. 2f). Finally, we found that fork density and origin firing rate in aneuploid cells were higher than euploid counterparts (Fig. 1n−p), suggesting that dormant replication origins were fired in the 1st S-phase following chromosome mis-segregation events. In our previous study we reported that aneuploid cells have reduced fork rate

and a higher number of stalled forks as compared to euploid cells[14]. Cells activate dormant origins in response to reduced fork rate and stalled forks to ensure that the genome gets fully replicated in time. Hence, we estimated origin firing rate and fork density, calculated as the total number of forks per Mb of DNA[24]. The total origin firing rate was twofold higher in aneuploid cells compared to euploid cells ($p = 1.5 \times 10^{-34}$). Consistent with the origin firing data, fork density was also 1.7-fold higher in aneuploid cells ($p = 7.46 \times 10^{-13}$). Analog-specific estimations for both the parameters also showed similar trends. Our data is also consistent with increased origin firing observed in aneuploid human pluripotent stem cells[25]. Overall, our data show that aneuploid cells struggle to complete replication and therefore

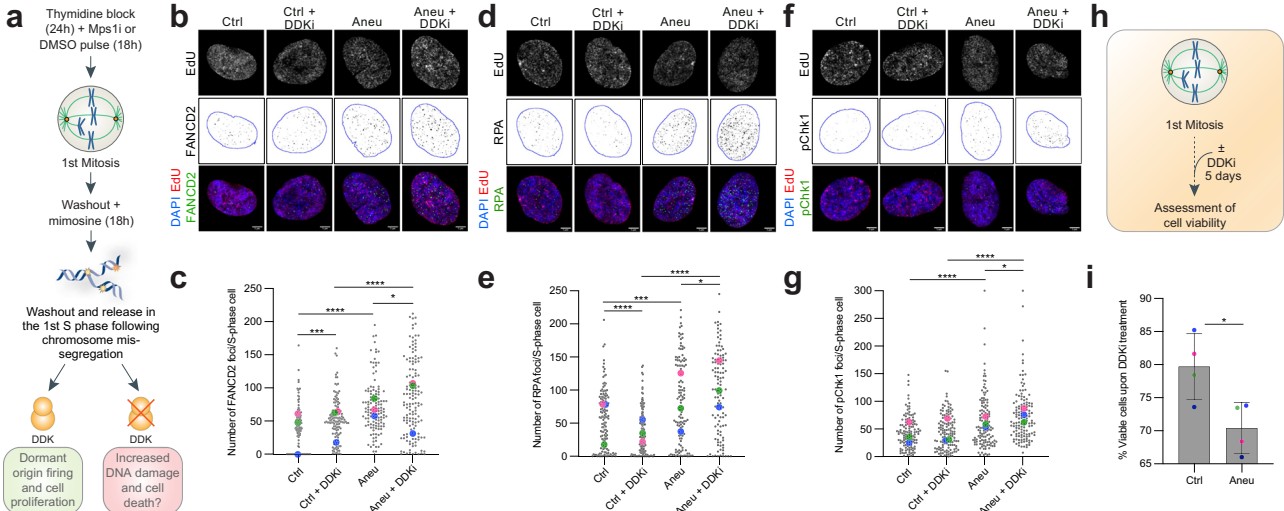

**Fig. 2 | DDK protects aneuploid cells from DNA damage accumulation and consequent cell death. a** Experimental workflow for the analysis of DNA replication stress markers in S-phase cells upon treatment with the DDK inhibitor XL-413. A short EdU pulse was performed before cell harvest in order to label S-phase cells. Representative images (**b**) and quantification (**c**) of FANCD2 foci per S-phase cell in control or aneuploid cells ±DDK inhibitor ($n = 123$ for Ctrl, $n = 125$ for Ctrl+DDKi, $n = 126$ for Aneu, $n = 137$ for Aneu+DDKi). * indicates $p = 0.0203$, *** indicates $p = 0.0002$ and **** indicates $p < 0.0001$. Representative images (**d**) and quantification (**e**) of RPA foci per S-phase cell in control or aneuploid cells ±DDK inhibitor ($n = 136$ for Ctrl, $n = 134$ for Ctrl+DDKi, $n = 119$ for Aneu, $n = 118$ for Aneu+DDKi). * indicates $= 0.0108$, *** indicates $p = 0.0001$ and **** indicates $p < 0.0001$. Representative images (**f**) and quantification (**g**) of pChk1 foci per S-phase cell in control or aneuploid cells ±DDK inhibitor ($n = 127$ for Ctrl, $n = 132$ for Ctrl+DDKi; $n = 131$ for Aneu; $n = 128$ for Aneu+DDKi). * indicates $p = 0.0449$ and **** indicates $p < 0.0001$.

**h** Experimental workflow for the assessment of cell viability upon exposure to the DDK inhibitor. **i** Quantification of live cells upon DDK inhibitor treatment in control and aneuploid cells (for each sample, values were normalized to the untreated control). $n = 4$ independent experiments. * indicates $p = 0.0251$. Ctrl, control (DMSO pulsed). Aneu, aneuploid cells (Mps1 inhibitor pulsed). DDKi, DDK inhibitor. Scale bars, 5 µm. LUT was inverted for FANCD2, RPA and pChk1 images. Blue borders in images are based on DAPI staining and define nuclei. Data are means of at least three biological replicates. Error bars in panel i represent SEMs. Two-tailed unpaired Student's $t$ test was performed for data in (**c**, **e**, **g** and **i**). In graphs, average values for each biological replicate are shown by colored dots (each color corresponds to a different biological replicate). Source Data are provided as a Source Data file. Drawings of schemes were made by partially utilizing extracts of figures published elsewhere[1].

activate backup mechanisms such as dormant origin firing to ensure genome duplication and tolerance to replication stress.

Altogether, these data provide crucial insights into the effects of aneuploidy on genome integrity. We find that: (1) cells harboring aneuploid karyotypes tend to accumulate increasing levels of chromosome abnormalities, (2) those defects might be the consequence of DNA replication stress, and (3) are correlated with a higher incidence of replication fork reversal and increased DNA damage markers. Finally, (4) at the same time, aneuploid cells also show an increased usage of dormant origins, which we speculate might act as a mechanism to tolerate aneuploidy-induced replication stress.

## Aneuploid cells rely on DDK to cope with replication stress

Dormant origin firing is a well-known rescue mechanism protecting cells during replication stress[26]. To test whether aneuploid cells would also rely on this salvage mechanism, we inhibited the activity of DDK, a key player in origin firing[27–31]. For this, aneuploid cells or pseudo-diploid counterparts (generated as in Fig. 1e) were arrested in late G1 (after the 1st mitosis) and then released in the presence or absence of the DDK inhibitor XL413[32] (Fig. 2a). After 6 h, cells were pulsed with the thymidine analogue ethynyl deoxy-uridine (EdU) for 30 min to label S-phase cells and then fixed and stained for FANCD2, RPA and pChk1. We found that inhibition of DDK led to significantly increased levels of FANCD2 (mean foci in control: $40.9 \pm 2.9$; control + DDKi: $52.6 \pm 2.9$; aneuploid $72.2 \pm 3.5$; aneuploid + DDKi: $84.5 \pm 4.9$), RPA (mean foci in control: $56.3 \pm 3.7$; control + DDKi: $36.2 \pm 2.9$; aneuploid $83.5 \pm 5.8$; aneuploid + DDKi: $103.9 \pm 6.4$) and pChk1 (mean foci in control: $44.1 \pm 2.6$; control + DDKi: $44.9 \pm 2.8$; aneuploid $62.9 \pm 4.2$; aneuploid + DDKi: $74.7 \pm 4.2$) in aneuploid cells, indicating that replication stress is exacerbated when interfering with dormant origin firing through DDK inhibition (Fig. 2b–g). These findings were validated also by using

alternative approaches to interfere with the function of DDK and Mpsi, namely an analog-sensitive allele of Cdc7[31] and the Mps1 inhibitor AZ3146[33], respectively (Supplementary Fig. 3a). For this, we inhibited Cdc7 using 20 µM 1-NM-PP1 analog - a concentration previously shown[31] to fully inhibit Cdc7-dependent phosphorylation of MCM2 (Supplementary Fig. 3b)- and induced aneuploidy using AZ3146. We found that Cdc7 inhibition resulted in significantly increased levels of FANCD2, RPA and pChk1 in aneuploid cells (Supplementary Fig. 3c–h), in agreement with experiments performed with XL-413. We also observed a significant increase in the number of FANCD2 foci and a significant decrease in that of RPA foci in control cells when Cdc7 was inhibited compared to functional Cdc7 (Supplementary Fig. 3c, d), which highlights the importance of DDK activity during replication stress and is in agreement with previous findings[31]. These results prompted us to test whether DDK activity, and its involvement in dormant origin firing, would also be critical for aneuploid cell proliferation. Interestingly, we found that aneuploid cells were more sensitive to DDK inhibition compared to pseudo-diploid counterparts (Fig. 2h, i)—although showing a similar rate of cell death under basal conditions (Supplementary Fig. 3i)-, indicating that they rely more than euploid cells on the function of DDK to survive. Altogether, our data show that DDK-mediated origin firing represents a protective mechanism that acts in S-phase of aneuploid cells to limit replication stress. Importantly, inhibition of this mechanism exacerbates replication stress in aneuploid cells and reduces their viability.

## Aneuploid cells undergo mitotic DNA synthesis to limit the consequences of replication stress on genome stability

DNA replication stress—defined as any slowing or stalling of replication fork progression and/or DNA synthesis[34]- impacts mitotic fidelity[35–37]. Thus, we thought to study how the events occurring in the 1st S-phase

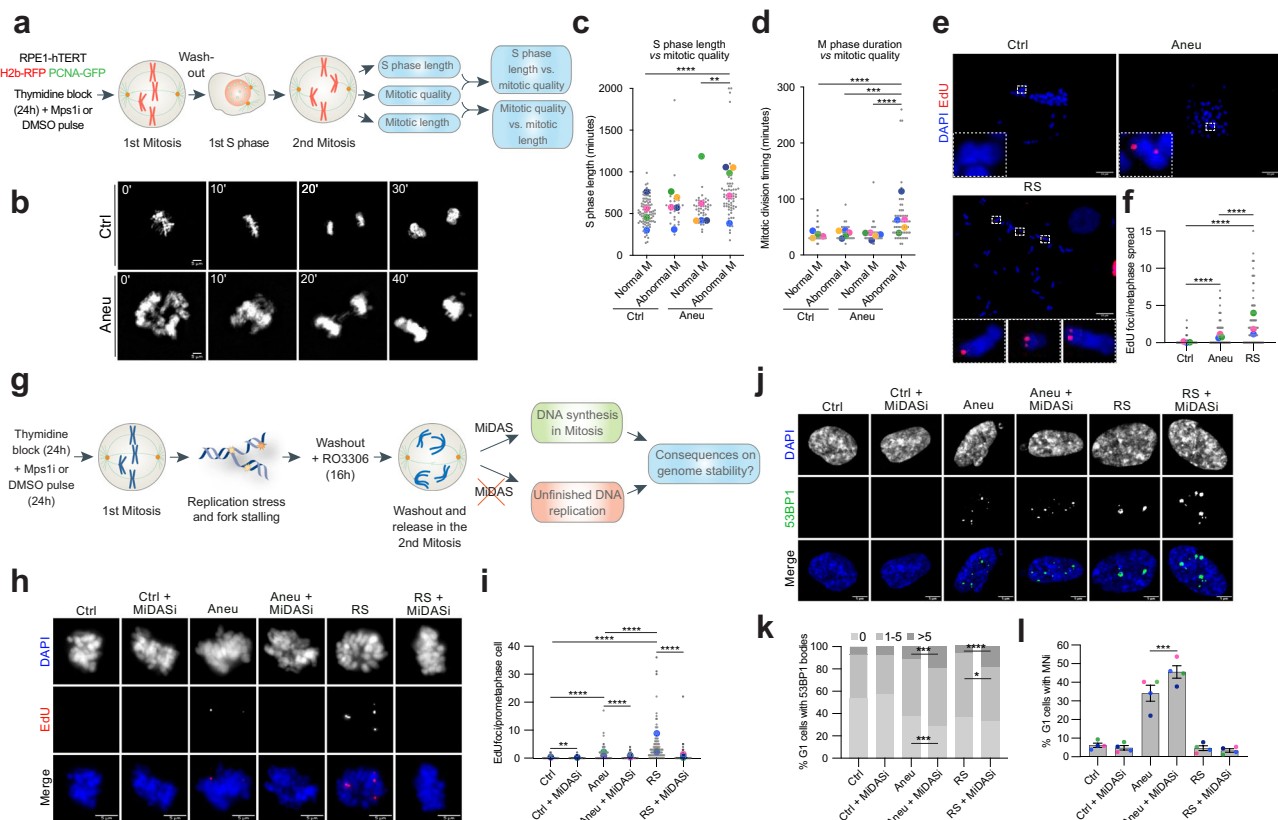

**Fig. 3 | MiDAS protects aneuploid cells from a further increase in their genome instability. a** Experimental workflow for the analysis of the 1st S-phase duration, 2nd M phase duration and quality by live-cell imaging in hTERT RPE-1 cells expressing H2b-RFP and PCNA-GFP. **b** Representative images from the movies of mitosis duration and quality in control and aneuploid cells. For illustration purposes, images were deconvoluted by using the Huygens software using the deconvolution express function. Correlation between (**c**) S-phase duration and quality of the subsequent mitosis and (**d**) mitotic timing and mitotic quality in control (for (**c**): $n = 98$ for Ctrl Normal M, $n = 27$ for Ctrl Abnormal M; for (**d**): $n = 98$ for Ctrl Normal M, $n = 34$ for Ctrl Abnormal M) and aneuploid (for (**c**): $n = 43$ for Aneu Normal M, $n = 63$ for Aneu Abnormal M; for (**d**): $n = 43$ for Aneu Normal M, $n = 62$ for Aneu Abnormal M) cells. Representative images (**e**) and quantification (**f**) of EdU incorporation on metaphase spreads in control ($n = 129$) and aneuploid ($n = 167$) cells. Cells treated with aphidicolin (RS, replication stress) were used as a positive control ($n = 106$). **** indicates $p < 0.0001$. **g** Experimental workflow for the assessment of genome instability in the following G1 phase upon MiDAS inhibition. Representative images (**h**) and quantification (**i**) of EdU incorporation in prometaphase cells upon MiDAS inhibition ($n = 130$ for Ctrl, $n = 113$ for Ctrl+MiDASi, $n = 156$ for Aneu, $n = 116$ for Aneu+MiDASi). Cells treated with aphidicolin (RS,

replication stress) were used as a positive control ($n = 111$ for RS, $n = 91$ for RS + MiDASi). ** indicates $p = 0.0081$ and **** indicates $p < 0.0001$. Representative images (**j**) and quantification of 53BP1 body (**k**) and micronucleus (**l**) accumulation in G1 cells following inhibition of MiDAS (for (**k**): $n = 544$ for Ctrl, $n = 518$ for Ctrl + MiDASi, $n = 543$ for Aneu, $n = 613$ for Aneu + MiDASi, $n = 446$ for RS, $n = 431$ for RS + MiDASi. For (**l**): $n = 556$ for Ctrl, $n = 643$ for Ctrl + MiDASi, $n = 543$ for Aneu, $n = 641$ for Aneu + MiDASi, $n = 397$ for RS, $n = 416$ for RS + MiDASi). In (**k**), * indicates $p = 0.0121$; *** indicates $p = 0,0007$ (Aneu vs Aneu + MiDASi with 0 53BP1 bodies) or $p = 0.0001$ (Aneu vs Aneu + MiDASi with >5 53BP1 bodies); **** indicates $p < 0,0001$. In (**l**), *** indicates $p = 0,0001$. Ctrl, control (DMSO pulsed). Aneu, aneuploid cells (Mps1 inhibitor pulsed). RS, replication stress (aphidicolin treated cells). MiDASi, MiDAS inhibitor. Normal M, normal mitosis. Abnormal M, abnormal mitosis. Scale bars, 5 or 10 μm. Data are means of at least three biological replicates. Error bars in (**l**) represent SEMs. Two-tailed unpaired Student's $t$ test was performed for data in (**c**, **d**, **f** and **i**). Two-sided Chi square test was performed for data in (**k**) and (**l**). In graphs, average values for each biological replicate are shown by colored dots (each color corresponds to a different biological replicate). Source Data are provided as a Source Data file. Drawings of schemes were made by partially utilizing extracts of figures published elsewhere[1].

of aneuploid cells affect the following cell division. To this aim, we performed live-cell imaging experiments with hTERT RPE-1 cells stably expressing PCNA-GFP and H2b-RFP. This allowed to monitor S-phase length through the measurement of time elapsed between appearance and disappearance of PCNA foci (a well-known feature of this DNA clamping factor[38]) and mitotic timing and quality by tracking chromosomes through H2b. Cells were synchronized with thymidine, then pulsed with reversine while they were transiting through the 1st mitosis. Cells were then washed-out and imaged every 10 min for 72 h in order to evaluate the duration of the first S-phase after chromosome mis-segregation and the quality of the 2nd mitosis (Fig. 3a, b). First, we found that aneuploid cells displayed a longer S-phase compared to euploid controls (mean S-phase length in control: $540.1 \pm 17.82$; aneuploid: $662.5 \pm 29.89$. Supplementary Fig. 4a), in agreement with the fact that they experience ongoing DNA replication stress and in line with previous reports[13,14]. Next, we decided to correlate S-phase length

to the quality of the 2nd mitosis. Thus, we classified mitotic figures in "normal mitoses", for those not displaying defects, and "abnormal mitoses", for those showing mitotic errors, including chromatin bridges, lagging chromosomes or micronuclei in the following G1. Interestingly, we found a positive correlation between S-phase length and frequency of abnormal mitoses (mean S-phase length in control: $603.3 \pm 55.4$; aneuploid: $728.7 \pm 46.2$) (Fig. 3c). Further, aneuploid cells that displayed mitotic errors spent more time in mitosis (Fig. 3d), which we could fully attribute to spindle-assembly checkpoint activation, since SAC inhibition rescued this delay (Supplementary Fig. 4b).

Based on the evidence that aneuploid cells suffer from replication stress in the 1st S-phase following chromosome mis-segregation events, we wanted to investigate whether they would attempt to finish DNA replication in the subsequent mitosis, as previously discovered in cancer cells as a consequence of S-phase stress[39]. In order to evaluate the activation of the mitotic DNA synthesis (MiDAS) pathway,

aneuploid cells generated as in Fig. 1a were arrested at the G2/M boundary with the CDK1 inhibitor RO3306 (Supplementary Fig. 4c, d—a synchronization method that did not induce DNA damage accumulation (Supplementary Fig. 4e–g)) and released in the presence of EdU and colcemid to monitor sites of active DNA synthesis in prometaphase cells. By doing so, we observed that the number of EdU foci per spread was significantly higher in aneuploid cells in comparison to the control (mean EdU foci in control: $0.1 \pm 0.04$; aneuploid: $0.9 \pm 0.1$; Fig. 3e, f). The DNA replication inhibitor aphidicolin was added in S-phase as replication stress inducer (mean EdU foci: $2.6 \pm 0.3$). Interestingly, we did not spot a tendency of MiDAS to occur at defined chromosomal locations such as telomeres, similarly to aphidicolin-induced MiDAS (Supplementary Fig. 4h, i). Next, to test the efficacy of MiDAS in fixing unfinished DNA replication, we inhibited the pathway and evaluated the consequences on genome stability in the following G1 (Fig. 3g). To this aim, we first tested if MiDAS could be inhibited by adding a high dose of aphidicolin in mitosis similarly as observed in cancer cells[39–41]. Our results showed that indeed this was the case, since the number of EdU foci per prometaphase cell was significantly reduced upon addition of aphidicolin in mitosis (mean EdU foci in control: $0.2 \pm 0.1$; control + MiDASi: $0.1 \pm 0.03$; aneuploid: $1.3 \pm 0.2$; aneuploid + MiDASi: $0.5 \pm 0.1$; Fig. 3h, i). As readouts of genome instability, we analyzed 53BP1 bodies and micronuclei in the G1 after the 2nd mitosis in which MiDAS had occurred. We found that both 53BP1 bodies per cell and the frequency of G1 cells with micronuclei were significantly increased in aneuploid cells in which MiDAS was inhibited in comparison with those in which MiDAS occurred properly (Fig. 3j–l). This correlation was also observed in aphidicolin-treated cells (mean EdU foci: $4.5 \pm 0.6$; + MiDASi: $0.7 \pm 0.3$), in agreement with previous studies[39]. These findings were further confirmed by depleting the non-catalytic subunit of DNA polymerase δ POLD3, which has been shown to be essential for the two MiDAS pathways described so far[39,41]. By using this approach and employing AZ3146 to induce aneuploidy (Supplementary Fig. 5a, b), we found that POLD3 depletion inhibited MiDAS (Supplementary Fig. 5c, d) and led to the accumulation of 53BP1 bodies and micronuclei in aneuploid cells (Supplementary Fig. 5e–g). Altogether, our data demonstrate that MiDAS acts as a safeguard mechanism in the 2nd mitosis to prevent genome instability from further increasing.

### Dormant origin firing and MiDAS protect aneuploid cells from further increase in genome instability

The results obtained from the characterization of the first S-phase after chromosome mis-segregation and the subsequent mitosis revealed two protective mechanisms operating in aneuploid cells with the role of limiting genome instability. To test if the combined action of these two pathways indeed serves to protect aneuploid cells, we simultaneously inhibited DDK in S-phase and MiDAS in the subsequent M phase and evaluated DNA damage and chromosomal aberrations in the following G1 phase (Fig. 4a). As DNA damage markers, we used FANCD2, a reliable replication stress/DNA damage marker in aneuploid cells (Fig. 1i and Supplementary Fig. 3c, d), and γH2AX, an early marker of DNA double-stranded breaks[42]. To specifically look at G1 cells, we used cytochalasin B to block cytokinesis and analyze daughter cells[43]. We observed that the number of both FANCD2 and γH2AX foci was significantly higher in aneuploid cells in which DDK and MiDAS were inhibited compared to aneuploid cells in which only either DDK or MiDAS was hindered (Fig. 4b–d). Interestingly, inhibition of both pathways led to an increase in FANCD2 and γH2AX foci also in euploid cells, highlighting that their proper functioning is crucial for maintaining genome integrity. Then, to assess the frequency of chromosomal aberrations upon DDK and MiDAS inhibition, G1 cells were treated with the PP1/PP2A phosphatase inhibitor calyculin A to induce premature DNA condensation[44] and obtain metaphase-like spreads. By mFISH analysis, we were able to observe an almost twofold increase in the percentage of cells with translocations between aneuploid cells in which DDK and MiDAS were or were not inhibited (mean percentage of cells with at least 1 translocation in aneuploid cells: 33.4; in aneuploid cells + DDKi + MiDASi: 63.2) (Fig. 4e–g). Taken together, these data indicate that the presence of both pathways protects aneuploid cells from further increasing their genome instability.

Interestingly, while scoring DNA damage in G1 cells we noticed that distribution of FANCD2 or γH2AX foci in aneuploid cells was not always symmetric between daughter cells. Thus, we specifically analyzed the pattern of DNA damage inheritance in the aneuploid sample and the euploid control, along with aphidicolin-treated cells where it has been recently shown that DNA damage can be distributed asymmetrically between daughters[45]. Our data indicated that FANCD2 and γH2AX foci were asymmetrically distributed in about 20 and 10% of aneuploid daughter cells, respectively (Fig. 4h–j). We then decided to follow the fate of aneuploid daughter cells via live cell imaging and asked whether asymmetric DNA damage distribution would impinge on proliferative capacity. For this, we used hTERT RPE-1 cells expressing H2b-GFP and RNF168-miRFP—as a marker of DNA damage- and assessed DNA damage inheritance in aneuploid cells over a 5-day timeframe (Supplementary Fig. 6a). First, in agreement with our findings obtained by monitoring FANCD2 and γH2AX in fixed cells, we found RNF168 to be asymmetrically partitioned in more than 20% of daughter cells (Supplementary Fig. 6b–d). Second, DNA damage transmission from mother to daughter cells was more likely to happen in aneuploid cells eventually displaying impaired proliferative capacity (defined as those cells that divided less than 4 times over a 5-day period (see "Methods")—Supplementary Fig. 6e), pointing at a correlation between DNA damage accumulation and reduced proliferative capacity. Lastly, and most importantly, the incidence of cell divisions in which DNA damage was asymmetrically distributed between daughter cells was higher in cells with low proliferative capacity (Supplementary Fig. 6f), indicating that unequal partitioning of DNA damage may act as an accelerator of cell cycle arrest in aneuploid cells. These data suggest that non-random distribution of DNA damage between aneuploid daughter cells, alone or in combination with other aneuploidy-associated features, such as micronucleation, proteotoxic stress, etc., could underlie the difference in proliferation observed among aneuploid cells[14,19].

### A method to separate arrested and cycling aneuploid cells

The asymmetric inheritance of DNA damage and cell fate determinants have been hypothesized to underlie stem cell self-renewal[46]. Thus, based on asymmetric portioning of DNA damage markers in aneuploid cells (Fig. 4h–j and Supplementary Fig. 6), we reasoned that, like stem cells, they might segregate DNA damage asymmetrically, partially explaining why some aneuploid cells can keep cycling while others get arrested and enter senescence[14,19]. To test this hypothesis, we first needed to confirm that a proportion of aneuploid cells indeed gets arrested in the cell cycle and becomes senescent over time. Hence, we let aneuploid cells progress for about 3 cell cycles before harvesting them for β-galactosidase staining, a widely used marker of senescence[47]. As a positive control, we used cells treated with doxorubicin for 7 days, as DNA damage is an established senescence-inducer[14]. Our results indicated that, as expected[14], there was a sub-population of senescent cells in the aneuploid sample (Fig. 5a). Also, by conducting the same experiment at earlier time points (i.e., 1, 2 or 3 cell cycles after aneuploidy induction), we found there was an accrual of senescent cells over time with a peak at 72 h after chromosome mis-segregation, reflecting the increase in genome instability as an important trigger of senescence in aneuploid cells (Supplementary Fig. 7a). To characterize, in detail, the aneuploid cells that were still able to cycle and those that underwent senescence, we decided to establish a method for their isolation and separation. For this, we reasoned that the main (and, at the same time, potentially exploitable)

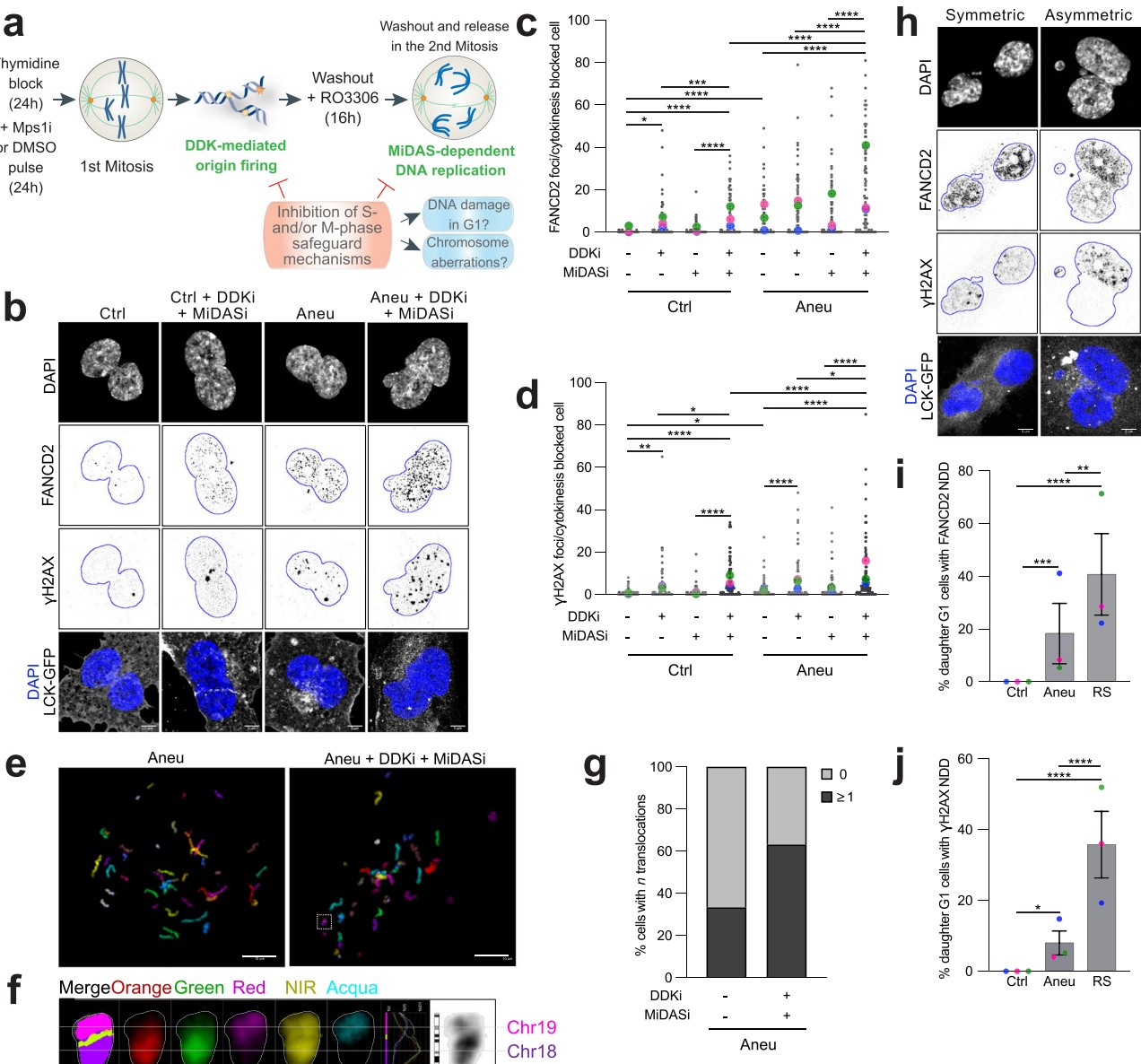

**Fig. 4 | DDK and MiDAS act as surveillance mechanisms to limit genome instability accumulation in aneuploid cells. a** Experimental workflow for the analysis of DNA damage and chromosome aberrations in G1 cells following DDK and MiDAS inhibition in the cell cycle after chromosome missegregation induction. Representative images (**b**) and quantification of FANCD2 (**c**) and γH2AX (**d**) accumulation in cytokinesis-blocked pseudo-G1 cells expressing LCK-GFP (for FANCD2: *n* = 90 for Ctrl, *n* = 72 for Ctrl+DDKi; *n* = 84 for Ctrl+MiDASi, *n* = 81 for Ctrl+DDKi +MiDASi; *n* = 87 for Aneu, *n* = 88 for Aneu+DDKi, *n* = 90 for Aneu+MiDASi, *n* = 86 for Aneu+DDKi+MiDASi. For γH2AX: *n* = 90 for Ctrl, *n* = 72 for Ctrl+DDKi; *n* = 84 for Ctrl +MiDASi; *n* = 81 for Ctrl+DDKi+MiDASi; *n* = 88 for Aneu, *n* = 93 for Aneu+DDKi, *n* = 95 for Aneu+MiDASi, *n* = 90 for Aneu+DDKi+MiDASi.). In (**c**), * indicates *p* = 0,0113, *** indicates *p* = 0.0003 and **** indicates *p* < 0,0001. In (**d**), * indicates *p* = 0,0122 (Ctrl vs Aneu) or *p* = 0.0152 (Ctrl+DDKi vs Ctrl+DDKi+MiDASi) or *p* = 0.0242 (Aneu+DDKi vs Aneu+DDKi+MiDASi); ** indicates *p* = 0.0055 and **** indicates *p* < 0,0001. **e** Representative mFISH images of G1 cell-derived metaphase-like chromosomes from aneuploid cells upon DDK and MiDAS inhibition. **f** Zoomed image of the chromosome highlighted in the dotted-line box in (**e**) (image on the right) from the aneuploid sample in which DDK and MiDAS were inhibited showing a translocation between chromosome 19 and chromosome 18. **g** Quantification of the percentage of cells with more than 1 translocation in the two samples (*n* = 18 for

Aneu, *n* = 19 for Aneu+DDKi+MiDASi). Representative images (**h**) and quantification of FANCD2 (**i**) and γH2AX (**j**) non-random distribution between the daughter pseudo-G1 cells (for FANCD2: *n* = 86 for Ctrl, *n* = 78 for Aneu. For γH2AX: *n* = 90 for Ctrl, *n* = 90 for Aneu). Cells treated with aphidicolin (RS, replication stress) were used as a positive control (*n* = 46 for FANCD2 and *n* = 81 for γH2AX). In (**i**), ** indicates *p* = 0.0021, *** indicates *p* = 0.0002 and **** indicates *p* < 0.0001. In (**j**), * indicates *p* = 0.0138 and **** indicates *p* < 0.0001. Ctrl, control (DMSO pulsed). Aneu, aneuploid cells (Mps1 inhibitor pulsed). RS, replication stress (aphidicolin treated cells). DDKi, DDK inhibitor. MiDASi, MiDAS inhibitor. NDD, non-random distribution. Scale bars, 5 or 10 μm. LUT was inverted for FANCD2 and γH2AX images. Blue borders in images are based on DAPI staining and define nuclei. Data are means of at least three biological replicates, except for data in (**e**–**g**) that were obtained from three biological replicates. Error bars in (**i**) and (**j**) represent SEMs. Two-tailed unpaired Student's *t* test was performed for data in (**c**) and (**d**). Two-sided Fisher's test was performed for data in (**i**) and (**j**). In graphs, average values for each biological replicate are shown by colored dots (each color corresponds to a different biological replicate). Source Data are provided as a Source Data file. Drawings of schemes were made by partially utilizing extracts of figures published elsewhere[1].

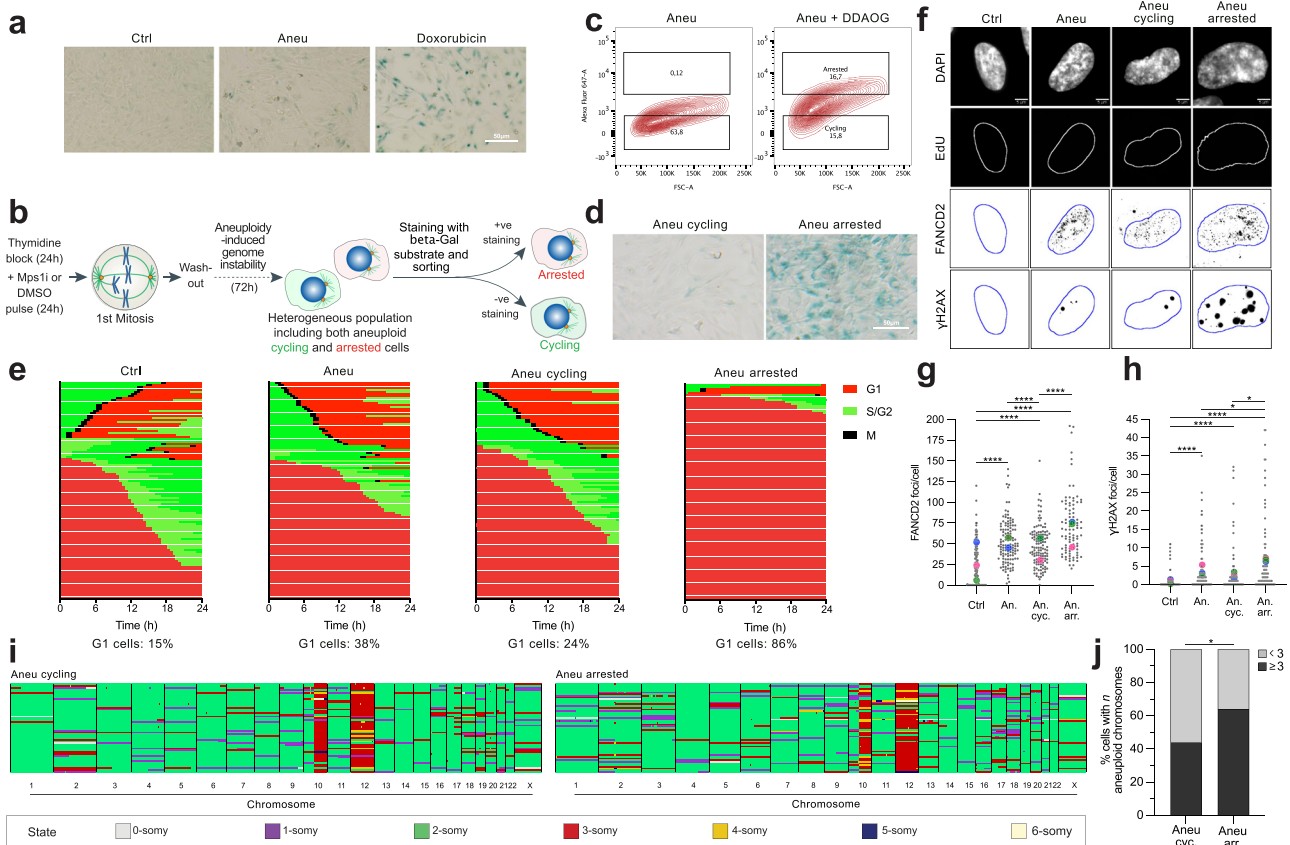

**Fig. 5 | Aneuploid cells that retained their proliferative capacity exhibit reduced levels of DNA damage and genome instability. a** Senescence-associated β-galactosidase staining in control and aneuploid cells. Doxorubicin-treated cells were used as a positive control. $N = 3$ independent experiments. **b** Experimental workflow for a method to separate and recover both cycling and arrested aneuploid cells based on FACS-sorting and the usage of a fluorescent substrate of the b-galactosidase enzyme. **c** FACS profiles showing the percentage of DDAOG-positive cells in aneuploid cells incubated or not with the DDAOG substrate. **d** Senescence-associated β-galactosidase staining in cycling and arrested aneuploid cells obtained after sorting. $N = 3$ independent experiments. **e** Cell cycle profiles of control, aneuploid, aneuploid cycling and aneuploid arrested FUCCI-cells analyzed by live-cell imaging and quantification of percentage of G1 cells in the four samples ($n = 100$ for each sample). Representative images (**f**) and quantification of FANCD2 (**g**) and γH2AX (**h**) foci per cell in the different samples. Only EdU negative cells were analyzed in order to exclude the contribution of S-phase cells present in the non-arrested cell samples ($n = 123$ for Ctrl, $n = 123$ for Aneu, $n = 123$ for Aneu cyc, $n = 89$ for Aneu arr). In (**g**) and (**h**), **** indicates $p < 0,0001$. In (**h**), * indicates $p = 0.0439$ (Aneu vs Aneu cycling) or $p = 0.0123$ (Aneu cycling vs Aneu arrested).

**I** scWGS of cycling and arrested aneuploid cells. Single cells are represented in rows and chromosomes are plotted as columns ($n = 50$ for Aneu cyc, $n = 53$ for Aneu arr). Copy-number states are indicated in colors (see legend at the bottom). hTERT RPE-1 cells have clonal gains of 10q and chromosome 12[73]. **J** Quantification of cells with at least 3 aneuploid chromosomes in the two samples ($n = 50$ for Aneu cyc, $n = 53$ for Aneu arr). Gains of 10q and chromosome 12 were excluded from the analysis. * indicates $p = 0.049$. Ctrl, control (DMSO pulsed). Aneu or An, aneuploid cells (Mps1 inhibitor pulsed). Aneu cycling or An cyc, aneuploid cycling cells. Aneu arrested or An arr, aneuploid arrested cells. Scale bars, 5 or 50 µm. LUT was inverted for FANCD2 and γH2AX images. Blue borders in images are based on DAPI staining and define nuclei. Data are means of three biological replicates, except for panel e (one replicate) and j (two replicates). Two-tailed unpaired Student's $t$ test was performed for data in (**g**) and (**h**). Two-sided Fisher's test was performed for data in (**j**). In graphs, average values for each biological replicate are shown by colored dots (each color corresponds to a different biological replicate). Source Data are provided as a Source Data file. Drawings of schemes were made by partially utilizing extracts of figures published elsewhere[1].

difference between aneuploid cycling and arrested cells is that the latter are senescent. Thus, we adapted to aneuploid cells a FACS sorting-based assay employing the fluorescent substrate of the β-galactosidase enzyme (which is highly active in senescent cells) 9H-(1,3-dichloro-9,9-dimethylacridin-2-one-7-yl)     β-d-galactopyranoside (DDAOG)[48]. We then exposed aneuploid cells to DDAOG and separated cells that were able to metabolize it (i.e., senescent cells) from those that could not metabolize it (i.e., cycling cells) (Fig. 5b, c and Supplementary Fig. 7b, c). First, we confirmed that sorted cells were indeed either arrested or cycling by β-galactosidase staining and found the former to be highly reactive to senescence-associated β-galactosidase staining (Fig. 5d). Further, we used hTERT FUCCI RPE-1 cells[49] (Supplementary Fig. 7d) to obtain the cell cycle profile of the two sorted aneuploid cell populations, together with the aneuploid sample before sorting and the euploid control. Our data confirmed that aneuploid cells positive for DDAOG were indeed arrested, since the

vast majority of them (86 ± 14.1%) were stuck in G1, as expected for senescent cells. On the other hand, negative ones were able to proliferate (Supplementary Fig. 7e and Fig. 5e) and the percentage of G1 cells was 24% (±8.5%) (Supplementary Fig. 7e and Fig. 5e). Collectively, these data indicate that our method allows for the successful separation and recovery of arrested and cycling aneuploid cells that could be used for further analysis and characterization of the two populations.

## DNA damage and karyotype complexity might be responsible, at least partially, for cell cycle defects of aneuploid cells

We then evaluated DNA damage in cycling and arrested aneuploid cells, as well as in aneuploid cells before sorting and the euploid control. Our data indicate that aneuploid arrested cells display increased levels of FANCD2 and γH2AX foci compared to aneuploid cycling cells (mean FANCD2 foci in aneuploid arrested: 71.8 ± 3.9; in

aneuploid cycling: 46.9 ± 2.2; mean γH2AX foci in aneuploid arrested: 6.4 ± 1.1; in aneuploid cycling 2.5 ± 0.5) (Fig. 5f–h). Further, we also analyzed the karyotype of cycling and arrested aneuploid cells by single-cell whole genome sequencing (scWGS) and observed an increased frequency of cells with at least 3 aneuploid chromosomes in arrested aneuploid cells (Fig. 5i, j) without showing any clear trend of specific gained/lost chromosomes (Supplementary Fig. 7f, g). Altogether, these data show that both DNA damage and severe karyotype imbalances might contribute, at least partially, to cell cycle arrest in aneuploid cells.

## Cycling aneuploid cells display increased expression of DNA repair genes

In line with reduced DNA damage and karyotype abnormalities in the aneuploid cells that retained their proliferation capacity, we found that the frequency of mitotic errors (such as anaphase bridges and micronuclei) in aneuploid cycling cells was comparable to that of controls for at least 3 generations (Fig. 6a–c). This result suggests that the karyotype of these cells is likely to remain stable over time, which is indicative of low levels of genome instability in aneuploid cycling cells. Having established a tool to separate the two sub-populations of aneuploid cells, we turned our attention to the identification of features distinguishing aneuploid cycling cells from those that arrested. To address this question, we decided to analyze their transcriptional signatures via RNAseq. This analysis revealed that the two samples are indeed quite different (Fig. 6d). In particular, aneuploid arrested cells displayed overexpression of p53 and inflammation-related genes, in agreement with previous findings[14]. Conversely, aneuploid cycling cells, as expected based on their retained ability to divide, exhibited increased expression of cell cycle genes compared to aneuploid arrested cells. Interestingly, we also noticed that DNA damage and repair genes were overexpressed in aneuploid cycling cells (Fig. 6d and Supplementary Table 1). Consistently, pathways associated with DNA damage repair were up-regulated in cycling vs arrested aneuploid cells (Fig. 6e). In line with this, when exposed to ionizing radiation (IR), cycling aneuploid cells were able to repair DNA damage more efficiently than the arrested ones (Fig. 6f, g and Supplementary Fig. 8a, b). Indeed, we measured a faster decay kinetics of γH2AX and 53BP1 levels in the cycling population compared to the arrested one (Fig. 6f, g and Supplementary Fig. 8a, b), that holds true for γH2AX also when compared to euploid controls (Supplementary Fig. 8c, d). Non-sorted aneuploid cells were also included in the analysis, and they turned out to have a lower efficiency compared to the euploid controls in fixing IR-induced DNA damage (Supplementary Fig. 8a, b, e, f). Increased expression of DNA damage repair genes found in aneuploid cycling cells could be due to their higher proliferative capacity compared to the arrested ones or the presence of euploid cells in the population (which were -1.8-fold more in cycling aneuploid cells compared to the arrested ones (24% vs. 13%, respectively)–Fig. 5i, j). To determine whether cycling aneuploid cells indeed upregulate DDR pathways, we compared their transcriptome to that of (cycling) euploid counterparts. As expected, when compared to euploid, cycling aneuploid cells downregulate pathways associated with cell proliferation and upregulate the p53 pathway (Supplementary Fig. 8g). Nonetheless, we found DNA repair gene expression to be higher in aneuploid cycling cells even when compared to euploid control cells (Supplementary Fig. 8h). As cycling aneuploid cells upregulate the DDR both in comparison to euploid cells and to arrested aneuploid cells, we conclude that this is not a consequence of higher proliferative capability and/or driven by a small fraction of diploid cells present in the population. Importantly, the activation of DNA damage repair pathways in aneuploid cells is consistent with our recent findings in cancer[20] and untransformed cells[50]. We therefore hypothesized that higher expression of DNA damage repair genes would confer a growth advantage to the cells. To confirm this, we turned to the CCLE (Cancer

Cell Line Encyclopedia) database[51,52] to analyze the association between doubling time and DNA damage repair gene expression in more than 400 human cancer cell lines. We divided the cell lines into top and bottom quartiles based on their doubling times, and then compared their gene expression profiles. Cell lines with a low doubling time (<35 h) exhibited increased expression of DNA repair related gene signatures in comparison to cells with a high doubling time (>65 h) (Fig. 6h–j), linking high levels of DDR with increased proliferation capacity. These data suggest that elevated expression of DDR genes might confer a proliferative advantage to aneuploid cells and make them able to cycle despite the disadvantage conferred by the aneuploid status. Altogether, our data reveal the existence of protective mechanisms in aneuploid cells, namely DDK-mediated origin firing in S-phase and MiDAS in the subsequent mitosis, which operate to limit their genome instability (Fig. 7). Also, increasing degrees of karyotype aberrations together with high levels of DNA damage could contribute, at least partially, to cell cycle arrest of aneuploid cells. Lastly, an increased capacity to repair DNA damage is associated with a proliferative advantage not only in untransformed but also in cancer cells (Fig. 7).

## Discussion
Genome instability is a hallmark of cancer[53]. Its most common form is CIN, shown to promote tumorigenesis and confer proliferative advantages to cancer cells[3–6]. CIN can directly cause aneuploidy and, in this study, we demonstrate that aneuploidy can also instigate CIN. By combining biochemical and live cell imaging experiments with single-molecule replication-mapping technologies and single-cell multi-omics analysis, we found that the acquisition of unbalanced karyotypes can directly contribute to short-term genome instability, which in turn yields a diverse array of karyotypic landscapes. This effect feeds a self-sustaining loop, in which aneuploidy leads to CIN, thus generating more aneuploid daughter cells able to propagate genome diversity through continuous errors during genome replication and segregation.

Previous reports have shown that aneuploid cells can experience replication stress[13,14]. Here, we show that a DDK-dependent dormant origin firing operates during the first S-phase following chromosome mis-segregation and acts as a protective mechanism to cope with replication stress. It is still unclear what are the actual sources of replication stress in aneuploid cells. DNA replication stress can be triggered by 1. collisions between the replication fork and the transcriptional machinery, 2. nucleotide pool imbalances, 3. difficulties in the template DNA (e.g., repetitive sequences and/or secondary structures), 4. scarcity of replication factors to perform DNA synthesis[34]. Among them, insufficient amount of replication factors seems to be the most likely cause of replication stress in aneuploid cells, based on the fact that decreased levels of MCM2-7 proteins were reported in RPE-1 and HCT116 stable aneuploid clones with defined trisomies[13]. Also, a recent study has revealed that tetraploid cells encounter replication stress as a result of insufficiency of DNA replication factors[54], a mechanism that could also apply to aneuploid cells. Because in our system we observe dormant origin firing, we speculate that the limiting DNA replication factors are those downstream of origin firing, such as PCNA, RFC and DNA polymerases[55]. Future studies will be aimed at exploring this possibility, with the goal of elucidating the contributions of those factors in DNA replication of aneuploid cells. This line of study might also open novel therapeutic interventions through selective targeting of aneuploid cancer cells by targeting those limiting DNA replication factors.

Along this line, another important implication of our finding that DDK and MiDAS play a central role in helping cells coping with aneuploidy is that these mechanisms could well be targetable vulnerabilities of aneuploid cancers. Although it might be challenging to

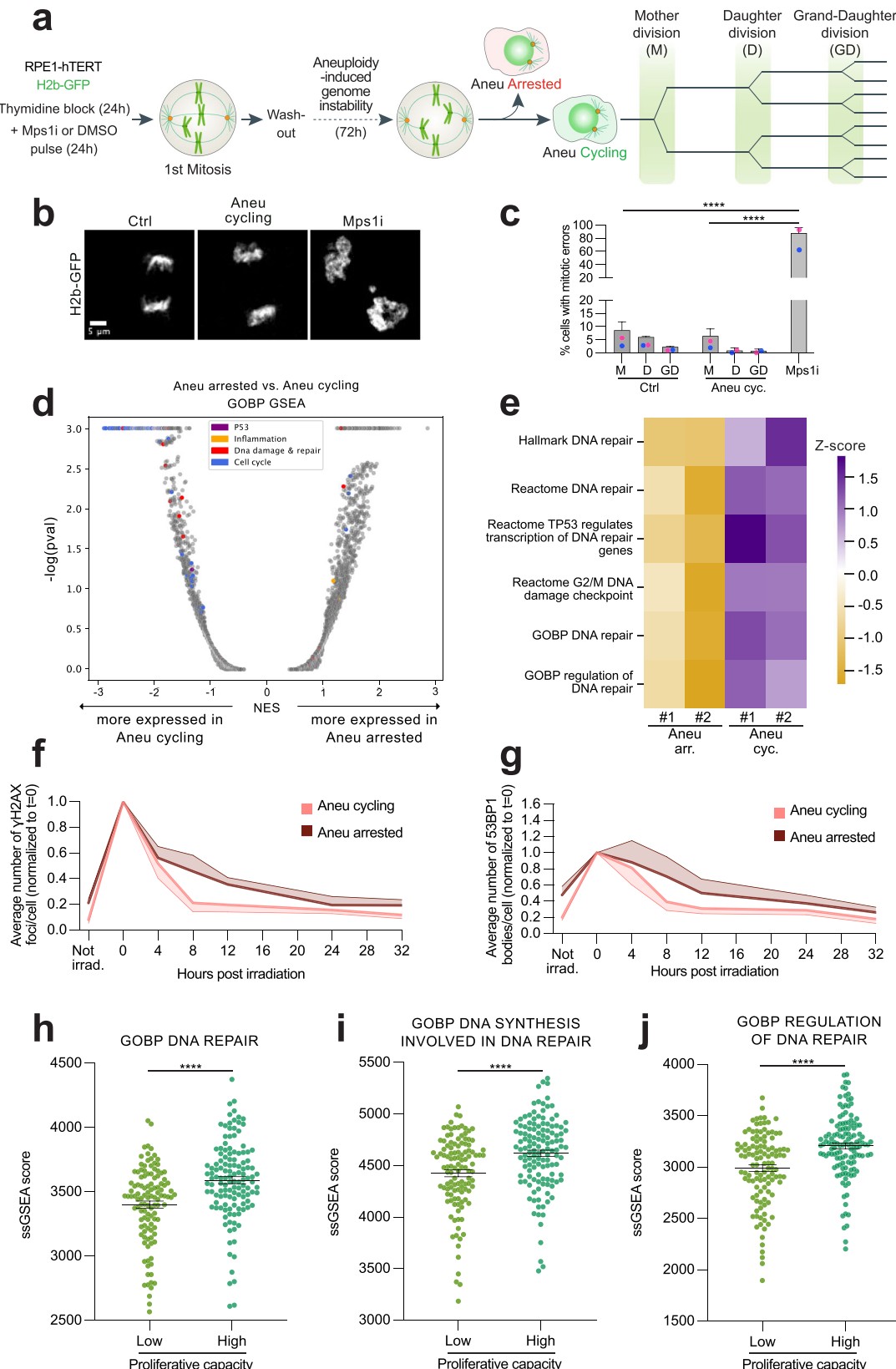

selectively target MiDAS - its key players SLX4-MUS81, RAD52 and POLD3[39,40] are also involved in other processes beyond MiDAS, such as homologous recombination (HR) repair and canonical S-phase DNA replication – things could be different for DDK. In particular, the DDK inhibitor TAK-931 was recently tested in a phase 2 clinical trial for the treatment of advanced solid tumors[56]. Based on our study, we

speculate that this drug (and similar ones) could be combined with inhibitors of DNA repair, such as the products of genes playing a role in HR repair, since they were the most differentially regulated in our analysis (Supplementary Table 1). Thus, the combined inhibition of DDK and HR repair factors could potentially be effective to treat aneuploid cancers. This approach would have the great advantage

**Fig. 6 | Cycling aneuploid cells display decreased karyotype aberrations and upregulate DNA repair genes in comparison with arrested aneuploid cells.** **a** Experimental workflow for the assessment of genome instability levels by live-cell imaging in aneuploid cycling cells expressing H2b-GFP. **b** Representative images of cell divisions in the different samples. For illustration purposes, images were deconvoluted by using the Huygens software using the deconvolution express function. **c** Quantification of mitotic errors in 3 cell division rounds in control (n = 187) and aneuploid cycling (n = 274) cells. Cells treated with the Mps1 inhibitor just before starting the time-lapse were used as a positive control for mitotic errors (n = 83). **** indicates p < 0.0001. **d** Volcano plot illustrating the differentially expressed pathways between cycling and arrested aneuploid cells. Specific gene sets are highlighted in color. Significance was determined with an empirical p value, calculated using 1000 permutations, using the default GSEA parameters. P values were adjusted for multiple testing using the FDR method. **e** Heat-map showing the z-scores of single-sample GSEA (ssGSEA) scores for DNA damage-related gene sets. Quantification of γH2AX foci (**f**) and 53BP1 bodies (**g**) in cycling (n = 150) and arrested aneuploid (n = 150) cells upon IR exposure. Only EdU negative cells were analyzed in order to exclude the contribution of S-phase cells present in the non-arrested cell samples. Association between ssGSEA score for GOBP DNA repair (**h**) or GOBP DNA synthesis involved in DNA repair (**i**) or GOBP regulation of DNA repair (**j**) and proliferation capacity in top vs. bottom quartiles of cancer cell lines from[51]. **** indicates p < 0.0001. Ctrl, control (DMSO pulsed). Aneu cycling or cyc, aneuploid cycling cells. Aneu arrested or arr, aneuploid arrested cells. M, mother division. D, daughter division. GD, grand-daughter division. Mps1i, Mps1 inhibitor. In (**e**), #1 and #2 refer to biological replicates. Not irrad., not irradiated. Data are means of two biological replicates, except for data in (**f**) and (**g**) (three replicates). Error bars represent SEMs. Shaded error bands in (**f**), (**g**) are shown above and below for arrested and cycling cells, respectively. Two-sided Fisher's test was performed for data in (**c**). Two-tailed paired t test was performed for data in (**f**) and (**g**). Two-tailed Mann–Whitney test was performed for data in (**h–j**). In (**c**), average values for each biological replicate are shown by colored dots (each color corresponds to a different biological replicate). Source Data are provided as a Source Data file. Drawings of schemes were made by partially utilizing extracts of figures published elsewhere[1].

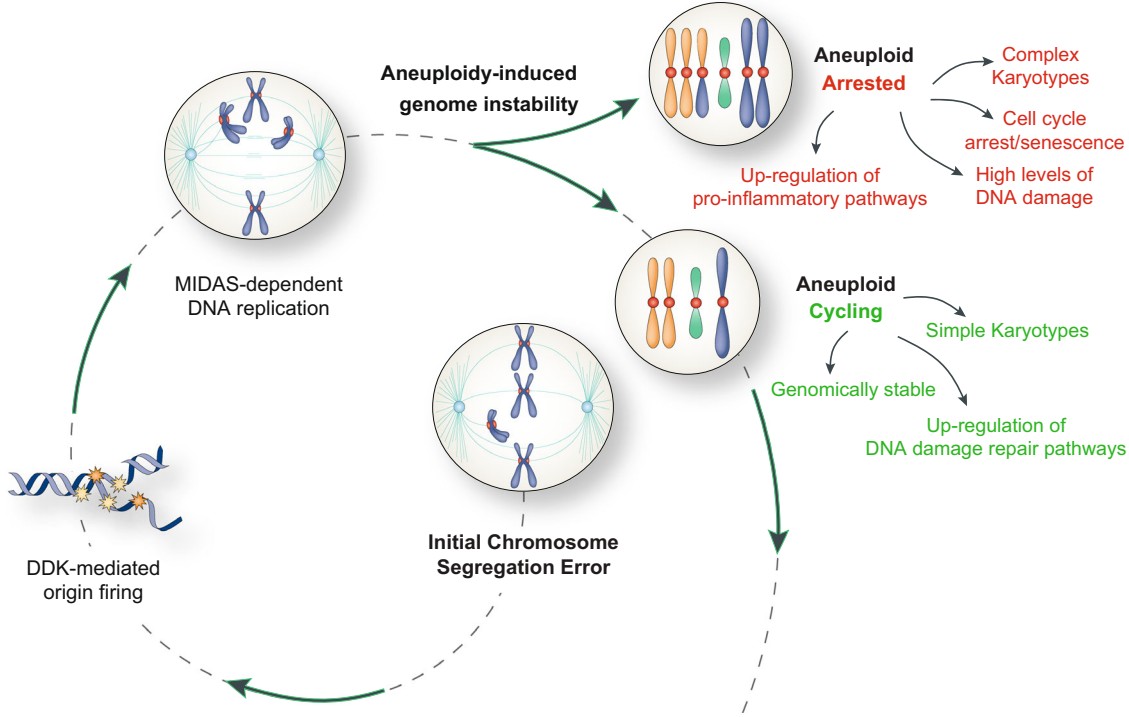

**Fig. 7 | Final model.** A model for how aneuploidy induces genome instability and its consequences. See text for more details. Drawings of schemes were made by partially utilizing extracts of figures published elsewhere[1].

of being highly selective against cancer aneuploid cells, lowering the side-effects of the DNA repair-based cancer therapies and the frequent chemoresistance associated with them[57].

In summary, by providing a detailed characterization of the short-term effects of unbalanced karyotypes in the acquisition of aggressive cancer-like features, we demonstrate that aneuploidy provides a point mutation-independent source of genome instability. Although this might offer a source of karyotypic variations capable of enabling proliferative capacity of cancer cells, it also leads to extensive DNA damage. Thus, we speculate that aneuploidy-induced genome instability might be a double-edged sword for cancer cells. On one hand, it is crucial for providing genome plasticity, on the other it might be extremely deleterious because of continuous DNA damage and replication stress. We propose that cancer cells solve this issue by limiting DNA damage - through upregulation of DNA repair genes - to a level compatible with cell proliferation. At the same time, this allows them to keep some degree of genomic instability and thus to continuously sample diverse karyotypic landscapes. Our observations

shed new light on the bidirectional association between aneuploidy and genomic instability and propose new approaches for the selective eradication of aneuploid tumors.

## Methods

### Cell culture conditions

The following hTERT RPE-1 cells were all tested free of mycoplasma contamination using Myco Alert (Lonza) according to manufacturer's instructions: hTERT RPE-1 cells, hTERT RPE-1 cells expressing H2b-RFP and PCNA-GFP or H2b-GFP or LCK-GFP (all generated in house), hTERT RPE-1 expressing PCNA-chromobodies and RNF168-miRFP (generated by Dr Simon Gemble and subsequently infected with H2b-GFP), Fluorescent Ubiquitination-based Cell Cycle Indicator (FUCCI) RPE-1 (kind gift of Professor Simona Polo, IFOM, Milan, Italy), hTERT RPE-1 expressing wild-type (wt) or analog-sensitive (as) Cdc7 (kind gift of Professor Prasad V. Jallepalli, MSKCC, New York, USA). All the cells were maintained in a humified environment at 37 degrees with 5% $CO_2$ and cultured in Dulbecco's modified Eagle's medium (DMEM) or

DMEM:F-12 (1:1 ratio) supplemented with 10% fetal bovine serum and 1% penicillin/streptomycin.

## Cell synchronization and treatments

To harvest cells for FACS analysis, hTERT RPE-1 cells were incubated with thymidine (5 mM, Merck Millipore) for 24 h or with mimosine (0,5 mM, Merck Millipore)/aphidicolin (400 nM, Merck Millipore)/RO3306 (7,5 µM Merck Millipore) for 18 h, washed 3 times in 1XPBS and harvested after 3,6, 12, 18 or 24 h.

To harvest metaphase cells for mFISH analysis, hTERT RPE-1 cells were synchronized at the G1/S boundary with thymidine, washed 3 times in 1XPBS (washout) and pulsed with the Mps1 inhibitor reversine (500 nM, Cayman Chemical) or the vehicle control (dimethyl sulfoxide, DMSO) for 24 h and then harvested for karyotype analysis either immediately after the pulse ('1st mitosis') or 24 h later ('2nd mitosis'). In both cases, colcemid (100 ng/ml, Merck Millipore) was added 2 h before harvesting the cells in order to block cells in prometaphase. To perform ultra-structural analysis of replication intermediates and analyze replication dynamics, cells were treated as above and analyzed immediately after the 24-h reversine/DMSO pulse.

To analyze DNA replication stress markers in S-phase, hTERT RPE-1 cells or hTERT RPE-1 cells expressing wt or as Cdc7 were plated onto fibronectin (5 µg/ml, Sigma-Aldrich) coated coverslips at approximately 30% confluence and synchronized with thymidine for 24 h. After washout, cells were pulsed for 18 h with reversine or AZ3146 (2 µM, Tocris) or vehicle control (DMSO) and released in the presence of mimosine for 18 h. After washout, cells were incubated for 6 h in the presence or absence of aphidicolin or the DDK inhibitor XL413 (10 µM, Aurogene) or the PP1 analog II 1-NM-PP1 (Cayman Chemical) prior to fixation for immunofluorescence. The thymidine analog ethynyl deoxy-uridine (EdU) was added at a final concentration of 10 µM during the last 30 min to label S-phase cells. To check levels of MCM2 phosphorylation in hTERT RPE-1 Cdc7 wt or as cells, after 24 h treatment with mimosine, cells were released and allowed to progress in S-phase in the presence or absence of 1-NM-PP1. 6 h later, they were harvested for Western blot analysis.

To assess viability in aneuploid and control cells, hTERT RPE-1 cells were synchronized with thymidine for 24 h, washed 3 times in 1XPBS and pulsed with reversine/DMSO for 24 h. Viability was assessed by FACS analysis 120 h after the reversine/DMSO washout. Cells treated with staurosporin (500 nM, Aurogene) for 24 h were used as a positive control for cell death.

To assess S-phase length and quality of the subsequent mitosis by time-lapse microscopy, after plating onto a fibronectin-coated glass 12wellplate, hTERT RPE-1 H2b-RFP and PCNA-GFP cells were blocked in G1/S with thymidine for 24 h, then washed out, pulsed with reversine/DMSO for 18 h and filmed for 72 h.

For MiDAS detection, hTERT RPE-1 cells were treated as above and, after reversine/DMSO washout, released in RO3306 for 12 h to arrest them in late G2 phase. Then, after 3 washes in 1XPBS, cells were released in mitosis in the presence of EdU 40 µM and colcemid for 2 h to harvest prometaphase cells for MiDAS detection. Aphidicolin 400 nM was added to a subpopulation of cells pulsed with DMSO just after DMSO washout in order to induce DNA replication stress (positive control). To investigate the consequences of MiDAS inhibition in G1 cells, cells were plated and treated as above. After RO3306 washout, cells were released in EdU 40 µM in the presence or absence of aphidicolin (2 µM, Merck Millipore) for 40 min (for prometaphase cells) or 3 h (for G1 cells) prior to fixation of immunofluorescence. For POLD3 knock-down experiments, hTERT RPE-1 cells were first transfected with POLD3 or scramble siRNAs and then re-plated onto fibronectin-coated coverslips and treated as above. An aliquot of the samples was harvested for Western blot analysis to check the levels of POLD3.

In order to harvest daughter pseudo-G1 cells, after thymidine and reversine pulses, hTERT RPE-1 LCK-GFP cells were released in the

presence or absence of XL413 10 µM and arrested in late G2 phase with RO3306. After drug washout, cells were incubated in the presence or absence of aphidicolin 2 µM for 3 h together with cytochalasin B (3 µg/ml, Merck Millipore) to block cytokinesis prior to fixation for immunofluorescence.

To obtain metaphase-like spreads from G1 cells, calyculin A (50 ng/ml, Sigma-Aldrich) was added for 45 min to G1 cells treated as above to induce premature chromosome condensation, prior to harvest for mFISH analysis.

To evaluate β-galactosidase positivity in aneuploid cells, after reversine washout, hTERT RPE-1 cells were allowed to progress for about 60 h before fixation. The same protocol was used to generate aneuploid cells for sorting, including hTERT FUCCI RPE-1 and hTERT RPE-1 H2b-GFP cells. After sorting, some cells were replated and fixed 16 h later to perform β-galactosidase staining, while some others were replated for live cell imaging experiments or directly harvested as frozen pellets for RNA extraction. As a positive control for senescence, cells continuously treated with doxorubicin (200 nM, Merck Millipore) for 7 days were used.

## Fluorescence-activated cell sorting (FACS) analysis

hTERT RPE-1 cells were treated as above, harvested, centrifuged and washed once in 1% bovine serum albumin (BSA) in 1XPBS. Cells were fixed in 100% ethanol (added dropwise while vortexing) and stored at +4 degrees overnight. Cells were centrifuged again and stained with staining solution -propidium iodide (20 µg/ml, Biotium), RNase A (1 mg/ml, Roche) in 1XPBS- for 3 h at room temperature (RT) prior to FACS analysis.

For assessing cell viability, hTERT RPE-1 cells were treated as above, harvested, centrifuged and resuspended in a Zombie Green™ Fixable Viability Kit (BioLegend) 1:100 in 1XPBS for at least 30 min at RT in the dark. Cells were centrifuged again and resuspended in 1XPBS.

In both cases, FACS was carried out on a FACS Celesta BD™ (laser 488, filter 695/40 nm for propidium iodide; laser 488, filter 530/30 nm for Zombie Green). For each condition, at least 20,000 events were recorded, and data analysis was performed using the FlowJo software (v10.7.2). Cells were gated for singlets (as shown in Supplementary Fig. 1b) and then, for Zombie Green stained cells, FITC intensity was used to distinguish live from dead cells (as shown in Supplementary Fig. 3i), i.e. the percentage of live cells was obtained subtracting debris and Zombie-positive cells (i.e. dead cells) from the total.

## Multicolor fluorescence in situ hybridization (mFISH)

After the treatments described above, hTERT RPE-1 cells blocked in prometaphase or G1 cells in which premature condensation was induced were trypsinized and centrifuged to obtain cell pellets. Cell pellets were resuspended in KCl 75 mM and incubated for 10 min in a 37 degrees waterbath. After centrifugation, cells were fixed in freshly-prepared Carnoy solution (methanol-acetic acid in a 3:1 ratio) while vortexing and then incubated for 30 min at RT. After a wash in freshly-prepared Carnoy solution, minimum volume of fixative was left to resuspend the pellet and cells were dropped onto glass slides. mFISH staining was performed following manufacturer's instructions (Meta-Systems). The Metafer imaging platform (MetaSystems) and the Isis software (MetaSystems, version 5.5) were used for automated acquisition of the chromosome spread and mFISH image analysis.

## Ultra-structural analysis of replication intermediates

hTERT RPE-1 cells were treated as above and harvested. Immediately after, cells were psoralen-crosslinked in vivo to stabilize replication intermediates as described in[58]. The cell suspension was first incubated with 30 µg/ml 4, 5′, 8-trimethylpsoralen (2 mg/ml, Sigma) for 5 min in the dark and then exposed to 365 nm UV light for 8 min in a UV Stratalinker 1800, (Stratagene), with 365 nm UV bulbs (model UVL-56, UVP) at 2–3 cm from the light source. The incubation and irradiation

steps were repeated three more times (4 cycles total). Genomic DNA (gDNA) was extracted with phenol-chloroform as described in[58]. 50 μg of gDNA were digested with KpnI and passed through a QIAGEN Genomic-tip 20/G column (QIAGEN) to enrich for replication intermediates, as described by Zellweger and Lopes[59]. EM spreads and imaging was performed as described in[60].

## Immunofluorescence analysis and EdU detection

At the end of the treatments described above, hTERT RPE-1, hTERT RPE-1 Cdc7 wt or as, or hTERT RPE-1 LCK-GFP cells were washed once in 1XPBS and then fixed in 4% paraformaldehyde (PFA) for 15 min at RT. After 3 washes in 1XPBS, cells were blocked in 3% BSA + 0.5% Triton-X in 1XPBS for 30 min and incubated with the following primary antibodies diluted in the same buffer for 90 min at RT: anti-FANCD2 (NB 100–182, Novus Biologicals) 1:200, anti-RPA (ab2175, Abcam) 1:200, anti-pChk1 (#133D3, Cell Signaling Technology) 1:50, anti-53BP1 (ab175933, Abcam) 1:1000, anti-γH2AX (JBW301, Millipore) 1:400. After 3 washes in 1XPBS, cells were incubated with secondary antibodies (Alexa-fluor 488 AffiniPure Donkey anti-Mouse IgG (H + L) Jackson ImmunoResearch 715-545-150; Alexa-fluor 488 AffiniPure Donkey anti-Rabbit IgG (H + L) Jackson ImmunoResearch 715-545-152; Alexa-fluor 647 AffiniPure Donkey anti-Mouse IgG (H + L) Jackson ImmunoResearch 715-605-150; Alexa-fluor 647 AffiniPure Donkey anti-Rabbit IgG (H + L) Jackson ImmunoResearch 715-605-152; Cy3 AffiniPure Donkey anti-Mouse IgG (H + L) Jackson ImmunoResearch 715-165-150; Cy3 AffiniPure Donkey anti-Rabbit IgG (H + L) Jackson ImmunoResearch 715-165-152. All used 1:400) diluted 1:400 in 3% BSA + 0.5% Triton-X in 1XPBS for 45 min at RT in the dark. Coverslips were then mounted on glass slides using Vectashield Antifade Mounting Medium with DAPI (Vectorlabs).

Where indicated, immunofluorescence was combined with EdU detection. Briefly, after the blocking, EdU detection was performed using the Click-iT EdU Cell Proliferation kit for Imaging (ThermoFisher Scientific) following the manufacturer's instructions. After the washes, incubation of the cells in primary antibodies and the subsequent steps of immunofluorescence were performed as above.

In all cases, images were acquired using Leica DM6B (Multi Fluo) or Leica SP5 microscope (63x objective was used in both cases).

## DNA fiber analysis

Cells were labeled sequentially with IdU (green [G]) and CldU (red [R]) and were harvested and processed as described in[14]. Data was collected from 2 independent experiments. A total of 47 and 54 Mb of DNA and 33 and 38 Mb of DNA was collected from the control and the aneuploid cells, respectively. Data analysis was performed as described in detail in[24]. Please note that the order of labeling is reversed (CldU→IdU) in the experiments described in[24], therefore, the interpretation of patterns is also reversed as compared to this article. Briefly, origin firing rate is the total number of origins that fired during the first and the second analog in each fiber divided by the total length of the un-replicated DNA in that fiber and the total length of the analog labeling pulses (120 min). Origins that fire during the first analog will appear as Red-Green-Red [RGR] and origins that fire during the second analog will appear as Red [R] events. However, origins that fire during the first analog will appear as RGR only if both the forks progress into the second analog. The origins will appear as RG or GR if either of the fork stalls or as G if both the forks stall. Thus, the total number of origins in each fiber was estimated by accounting for the probability of forks stalling.

Fork density is the total number of forks in each fiber divided by the total length of the un-replicated DNA of that fiber. Origins and termination events account for 2 forks each and unidirectional fork events account for 1 fork each. However, some of the unidirectional forks could be an origin whose left or rightward fork is stalled. Thus, the total number of forks on each fiber was estimated by accounting

for the probability of forks stalling. Please see[24] for calculation of fork stall rate and how the probability of fork stalling was used to estimate the final origin firing rate and fork density for each fiber.

## Cell proliferation assay

After thymidine synchronization and 18 h reversine/DMSO pulse as above, hTERT RPE-1 cells were trypsinized, counted and plated into a 96wellplate in the presence of XL413 10 μM or the vehicle control (DMSO). Drugs were re-added fresh every 48–72 h during the 120 h-treatment. Then, cell viability was assessed by using the CellTiter-Glo Luminescent cell viability assay (Promega) following manufacturer's instructions.

## Western blot analysis

hTERT RPE-1 Cdc7 wt or as cells or hTERT RPE-1 cells treated as above were harvested and lysed using RIPA buffer (1x, Cell Signaling Technology) supplemented with protease inhibitor cocktail (Roche) and phosphatase inhibitor cocktail (Merck Millipore) for 15 min on ice. After sonication, whole cell lysates were quantified and aliquots of samples were resolved on a Criterion TGX Stain-Free precast gel (Bio-Rad). The membrane was blocked in 1XPBS + 0,1% Tween (1XPBS-T) with non-fat dry milk (5%, Merck Millipore) for 30 min at RT and then incubated for 3 h at RT with a primary antibody resuspended in 1XPBS-T with 5% milk. Following 3 × 10 min washes using 1XPBS-T, the membrane was incubated for 45 min at RT with a HRP-conjugated secondary antibody (anti-mouse Cat# P0447 Agilent 1:5000 or anti-rabbit Cat# P0448 Agilent 1:5000) re-suspended in 1XPBS-T followed by 3 × 10 min washes in PBS-T. Clarity Western ECL Substrate (Bio-Rad) was used for signal detection, and images were captured on a ChemiDoc XRS + (Bio-rad). Primary antibodies used were: MCM2 (1:1000, A300-191A, Bethyl Laboratories), phosphoS40MCM2 (1:1000, ab133243, Abcam), POLD3 (1:1000, ab182564, Abcam), GAPDH (1:2000, #2118, Cell Signaling Technology) and vinculin (1:5000, V9131, Merck Millipore).

## Live cell imaging

To monitor S-phase length and M phase duration and quality, hTERT RPE-1 cells were treated as above. After reversine/DMSO washout, fresh medium without phenol red was added to the cells. Cells were imaged every 10 min for 72 h using a 20x/0.75 NA objective with an inverted Nikon Eclipse Ti microscope equipped with an incubator for live cell imaging. The same microscope and acquisition settings were used to film hTERT RPE-1 H2b-GFP aneuploid cycling cells after sorting for 72 h to assess their genome stability and to film hTERT RPE-1 expressing PCNA-RFP, RNF168-miRFP and H2b-GFP, with the only exception that in the latter case cells were imaged for 110 h (only miRFP and GFP channels were acquired).

## MiDAS detection

To detect MiDAS on metaphase spreads, after incubation in EdU and colcemid as above, cells were treated similarly to those for mFISH analysis. After cell dropping onto glass slides and complete evaporation of the Carnoy solution, slides washed in 1XPBS in agitation. EdU detection was performed with the Click-iT EdU Cell Proliferation kit for Imaging (ThermoFisher Scientific) according to manufacturer's instructions with some minor modifications as described in[61]. Slides were then mounted using Vectashield Antifade Mounting Medium with DAPI.

To detect MiDAS on prometaphase cells, at the end of the treatments described above, cells were fixed in 4% PFA for 15 min at RT, then washed 3 times in 1XPBS. EdU detection was performed as above[61] and coverslips were then mounted on glass slides using the same mounting medium as above.

In both cases, images were acquired using Leica DM6B (Multi Fluo) or Leica SP5 microscope (63x objective was used in both cases).

## RNA interference

hTERT RPE-1 cells were transfected with POLD3 (ThermoFisher, #4390824) or non-targeting (Dharmacon) smartPool siRNAs at a final concentration of 80 nM by using Lipofectamine RNAiMax transfection reagent (Thermo Fisher Scientific) according to the manufacturer's instructions.

## Telomeric FISH

A Cy3-labeled, C-rich PNA telomere probe (Panagene) was used for telomeric FISH on metaphase spreads according to manufacturer's instructions with minor modifications. At the end of the FISH protocol, EdU detection was performed as indicated above. Slides were mounted as above and images were acquired using Leica DM6B (Multi Fluo) microscope (63X objective).

## Analysis of distribution of DNA damage in fixed-pseudo-G1 cells

First, the number of FANCD2 and γH2AX foci per daughter cell was counted. Based on the average number of foci per cell, cells with less than 6 or 4 foci (for FANCD2 or γH2AX, respectively) were excluded from the analysis. Then, the total number of foci between the two daughters (e.g. 13 + 10) was calculated and divided by 2 to obtain the number of foci predicted to be inherited by each daughter cell in case of symmetric distribution of DNA damage (e.g. 23 divided by 2 is equal to 11.5). Subtraction or addition of this number to the total number of foci in the daughters was used to set a threshold for asymmetric distribution of DNA damage, e.g. 23–11, 5 = 11.5, which is the lowest threshold; 23 + 11.5 = 34.5, which is the highest threshold; if a daughter cell has less than 11.5 or more than 34.5 foci the distribution of DNA damage is considered as non-random (non-random distribution, NDD).

## Analysis of distribution of DNA damage in live G1 cells

Transmission of DNA damage and its distribution pattern at each cell division were assessed in newly-born G1 hTERT RPE-1 cells expressing PCNA-chromobodies, RNF168-miRFP and H2b-GFP. RNF168 foci were counted 12–24 frames after the end of cell division, *i.e.* when DNA started to decondense, which correspond to 2–4 h in the G1 phase. First, the frequency of cell divisions with DNA damage transmission was calculated as the percentage of divisions in which daughter cells inherit DNA damage over the total number of cell divisions, independently of its distribution pattern. Second, the frequency of cell divisions in which DNA damage was asymmetrically partitioned was calculated as the percentage of divisions in which daughter cells inherit an unequal number of RNF168 foci (typically 1:0 or 2:1 split) over the total number of cell divisions. Because cells dividing approximately every 24 h would divide about 5 times during an almost 120 h-timelapse experiment and aneuploid cells are known to cycle at a slower rate, a cut-off at 4 cell divisions was set in order to distinguish cells with different proliferative capacity: cells undergoing less than 4 mitoses would be classified as cells with 'low' proliferative capacity, while cells undergoing at least 4 mitoses would be named as cells with 'high' proliferative capacity.

## β-galactosidase staining

DMSO- and reversine-pulsed cells allowed to progress for either 24, 48 or 72 h after DMSO/reversine washout, together with sorted cycling and arrested aneuploid cells (when indicated), were plated into a 6 well plate at $1 \times 10^6$ cells/well and allowed to attach overnight. Then, cells were stained using the Senescence β-Galactosidase Staining Kit (Cell Signaling Technology) following manufacturer's instructions and images were acquired using the EVOS imaging system (Thermo Fisher Scientific).

## Sorting of aneuploid cells

Cells were plated into 150 mm plates (900.000 cells/plate) and treated as above. After reversine/DMSO washout, cells were allowed to divide for about 60 h. Then, they were incubated with the fluorescent substrate of the β-galactosidase enzyme DDAO-Galactoside (DDAOG) 10 mM for 90 min. At the end of the incubation, cells were harvested for FACS sorting and acquired using a FACSAria Fusion flow cytometer (BD). Cells were first gated for singlets and alive cells (based on DAPI staining); then, FSC-A and Alexa-647 intensity were used to distinguish cycling from arrested cells, i.e. cycling cells were gated using the same Alexa-647 Mean Fluorescence Intensity of the control sample without DDAOG, while arrested cells were gated imposing 0,1% or 0,5% on Alexa-647 signal to the control sample without DDAOG. FlowJo was used to perform data analysis and generate the plots in Fig. 5 and Supplementary Fig. 7.

## Cell cycle profile analysis through the FUCCI system

hTERT FUCCI RPE-1 cells were used to generate euploid, aneuploid, cycling aneuploid and arrested aneuploid cells as described above. After the sorting, cells were plated in a 12wellplate, allowed to attach overnight and then filmed using an inverted Nikon Eclipse Ti microscope with a 20x/0.75 NA objective equipped with an incubator for live cell imaging. Brightfield, green (GFP) and red (mCherry) channels were used to acquire the movie. Images were taken every 30 min for 24 h. Cell cycle stage was determined based on nuclear color: red nuclei were scored as G1 phase, while yellow and green nuclei were scored as S/G2 phase; lastly, M phase was characterized by uncolored nuclei of two dividing cells[49].

## Sample processing for RNAseq

Aneuploid cycling and arrested cells post sorting were centrifuged and cell pellets were obtained. RNA was extracted from them using a RNeasy kit (QIAGEN) and its quality was assessed with a Bioanalyzer 2100 (Agilent). Then, for each sample, total RNA was depleted of ribosomal RNA and the RNAseq libraries were prepared with the Illumina TruSeq Stranded Total RNA kit following the manufacturer's protocol. Briefly, after the fragmentation of RNA using divalent cations at elevated temperature, cDNA was synthesized, end-repaired and 3′-end-adenylated. Following adapter ligation, libraries were amplified by PCR. Amplified libraries were checked on a Bioanalyzer 2100 (Agilent) and quantified with picogreen reagent. Libraries with distinct TruSeq adapter UDIndexes were sequenced for 50 bases in the paired-end mode with 35 million reads in coverage on a Novaseq 6000 sequencer.

## Data analysis of RNAseq

RNA reads were aligned to the GRCh38 primary assembly with Ensembl 104[62] gene annotations using. STAR 2.7.9a[63]. Gene counts were quantified with subread 2.0.2[64]. Differentially expressed genes were determined using DESeq2 1.30.0[65] with a Wald test, regressing out for the batch factor. Subsequently, genes were filtered based on significance (*p*-value δ0.05 and pADJ δ0.25; for the aneuploid *vs.* control analysis results were filtered only based on *p*-value). Pre-ranked gene set enrichment analysis (PreRanked GSEA) was performed to identify enriched pathways[66]. Single cell gene set enrichment analysis (ssGSEA) was performed using GenePatterns[66,67], to compare the expression of specific DNA damage-related signatures across samples. Z-scores were calculated for each gene signature across and plotted as a heatmap. Plots were generated using the Python's 'seaborn' library (Van Rossum and Drake, 2009).

## Kinetics of DNA repair upon exposure to IR

After cell sorting (as described above), aneuploid cycling and arrested cells, together with non-sorted aneuploid cells and euploid controls, were plated on coverslips and incubated overnight. The day after, cells were γ-irradiated (1.25 Gy) with the Faxitron CP-160 (Faxitron x-ray Corporation) and fixed for immunofluorescence as indicated above at different timepoints. In order to exclude S-phase cells, cells were

pulsed with EdU 10 µM for 30 min before fixation. Non-irradiated cells were fixed together with the first timepoint. i.e. 0 h post irradiation. For each sample, the number of γH2AX foci/53BP1 bodies per cell was counted, their average per cell was calculated and normalized to that at 0 h post irradiation.

## Association of DDR gene expression with doubling time

CCLE (Cancer cell line encyclopedia) gene expression data were obtained from DepMap [https://depmap.org/portal/] 22Q1 release[52] and cell line doubling times were obtained from[51]. Cell lines were divided to quartiles according to their doubling time, while ssGSEA scores were generated using the GenePattern platform [https://www.genepattern.org/][66,67] and compared between the top and bottom quartiles. Statistical analysis (two-tailed Student' t test) and plotting were performed using GraphPad PRISM v9.3.1.

## Sample processing for single-cell whole genome sequencing (scWGS)

Cell pellets were resuspended in cell lysis buffer (100 mM Tris-HCl pH 7.4, 154 mM NaCl, 1 mM CaCl$_2$, 500 µM MgCl$_2$, 0,2% BSA, 0,1% NP-40, 10 µg/ml Hoechst 33358, 2 µg/ml propidium iodide in ultra-pure water) and incubated on ice in the dark for 15 min to ensure complete lysis. Resulting single nuclei of G1 phase (as determined by Hoechst and PI staining) were sorted into single wells of 96 wellplates on a MoFlo Astrios cell sorter (Beckman Coulter) and sorted at −80 degrees until further processing. Automated library preparation was performed (Bravo Automated Liquid Handling Platform, Agilent Technologies) as previously described[68]. Resulting single-cell libraries were pooled for subsequent sequencing.

## Data analysis for scWGS

Sequencing was performed using a NextSeq 500 machine (Illumina; up to 77 or 68 cycles – single end; excluding sample-specific barcodes). Reads were afterwards aligned to the human reference genome (GRCh38/hg38) using Bowtie2 (version 2.2.4 or 2.3.4.1[69];). Duplicate reads were marked with BamUtil (version 1.0.3;[70]) or Samtools markdup (version 1.9[71]. The aligned read data (bam files) were analyzed with a copy number calling algorithm called AneuFinder [https://github.com/ataudt/aneufinder][72]. Following GC correction and blacklisting of artefact-prone regions (extreme low or high coverage in control samples), libraries were analyzed using the dnacopy and edivisive copy number calling algorithms with variable width bins (average bin size = 1 Mb; step size = 500 kb). Results were afterwards curated by requiring a minimum concordance of 95% between the results of the two algorithms. Libraries with on average less than 10 reads per bin (~ 55,000 reads for a diploid genome) were discarded. A chromosome was classified as aneuploid when at least 95 % of the bins showed a deviation from euploid (deviation from 2-somy). Chromosomes 10 and 12 were excluded for the calculation of whole-genome scores.

## Image analysis

After acquisition, microscopy images were analyzed with Fiji (version 2.9.0) or Arivis software (version 3.6.0.) and processed with Fiji or Huygens software (version 19.04). Details of image processing, where applicable, are annotated in figure legends.

## Quantification and statistical analysis

All statistical analyses were performed using GraphPad Prism software (9.2.0 version). Statistical significance in each case was calculated using two-tailed Student's t test, two-sided Chi-squared test or Fisher's exact test. Where not indicated on the graphs, differences were not statistically significant. The details of error bars, number of biological replicates and n for each experiment are annotated in figure legends.

## Data availability

RNAseq and scWGS data generated during this study have been deposited at the EMBL-EBI repositories ENA and ArrayExpress with the following accession numbers E-MTAB-12537 and PRJEB58407, respectively. The remaining data are available within the Article, Supplementary Information or Source Data files. Source data are provided with this paper.

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

## Acknowledgements

We are very grateful to all the members of the Santaguida lab for fruitful discussions throughout the project. We thank Elsa Logarinho and members of M. Mapelli lab for constructive discussions, and Elia Zanella for help with telomeric FISH. We acknowledge help from Flow Cytometry, Genomic and Imaging units at IEO for their technical assistance. Work in the Santaguida lab is supported by grants from the Italian Association for Cancer Research (MFAG 2018 - ID. 21665 project), Ricerca Finalizzata (GR-2018-12367077), Fondazione Cariplo, the Rita-Levi Montalcini program from MIUR and the Italian Ministry of Health with Ricerca Corrente and 5×1000 funds. L.G. is supported by a fellowship from Fondazione IEO-Monzino. M.R.I. is supported by an AIRC Fellowship (ID 26738-2021). Work in the Ben-David lab is supported by grants from the European Research Council (ERC grant #945674), Israel Cancer Research Foundation Gesher Award, Israel Science Foundation (ISF grant #1805/21), the DoD CDMRP Career Development Award (grant #CA191148), the Azrieli Faculty Fellowship and the EMBO Young Investigator Program. D.F. receives salary support from the CNRS and I. Curie, M.D. from I. Curie. Work in Fachinetti lab was supported by an Emergence Grant 2018 from the City of Paris.

## Author contributions

L.G. and G.D.F. performed the vast majority of the experiments, analyzed the data and prepared the figures. V.M. performed the experiments presented in Fig. 5a–f and contributed to the experiments presented in Supplementary Figs. 1a, b, 2a–f, 3b, i, 4c, d, 7b, e. M.G. and Y.D. generated the EM data. M.D. and D.F. performed the mFISH analyses. Y.E. and U.B.D. analyzed the RNAseq data and CCLE data. R.W., A.E.T., D.C.J.S., M.S. and F.F. performed the scWGS and RNAseq. MRI performed the viability assay in Fig. 2i. DRI and N.R. performed the DNA fiber analysis. S.T. generated the data presented in Supplementary Fig. 7a. C.S. assisted with live cell imaging experiments. S.G. and R.B. provided the hTERT RPE-1 expressing PCNA-chromobodies and RNF168-miRFP. S.S. conceptualized the study, coordinated the research and wrote the manuscript together with L.G. and G.D.F. and with input from all the authors.

## Competing interests

The authors declare no competing interests.
