## [Peer Review File · Nature Communications]

REVIEWER COMMENTS

Reviewer #1 (Remarks to the Author):

The manuscript by De Feudis et al., provides interesting novel insights into the mechanisms of short-term genome instability following induction of chromosome mis-segregation and aneuploidy. By focusing on the 1st and second cycle after missegregation, the authors offer a high temporal resolution on the underlying mechanism of how aneuploidy may induce genomic instability. For publication, the paper would benefit from additional experiments and edits on the text to strengthen the conclusions and discuss the remaining open questions (i.e. what the study does not address, such as the long term consequences of CIN/aneuploidy).

Major concerns

1. A general concern of several results shown in the paper is the fact that many analyses rely on using drugs/chemicals to inhibit genes or pathways and in most cases are not validated by genetic approaches (based on CRISPRi or cDNA overexpression). While the time sensitive experiments based on reversine may be tricky to recapitulate genetically, they should be repeated and validated using alternative MPS1 inhibitors. For the other experiments, including inhibitors of DDK and MiDAs, the key experiments (especially Fig. 2 and 3) should be confirmed using a genetic approach.
2. Examining the short-term effects of aneuploidy on genome integrity is an important question in the field. However, it is also very difficult to distinguish the effect of CIN and aneuploidy. In figure 1, the authors induce aneuploidy by pulsing the cells with reversine which induces the missegregation of chromosomes and then compare the cells from the 1st mitosis and 2nd mitosis. The problem is the following: are the observed karyotypes/cellular changes (or other difference) observed in 2nd mitosis the consequence of aneuploidy in the 1st mitosis or of long-term disruption of chromosome stability by reversine? In other words, can reversine impair chromosome stability in a long-term and aneuploidy-independent manner even though it had been washed out? The authors should discuss this potential caveat.
3. As the authors report in figure 4d and related panels, inhibition of DDK and MiDAS can lead to an increase in FANCD2 and gamma-H2AX foci in euploid cells. This means those two phenotypes are not specific to aneuploid cells. This raises two questions: (1) why does the inhibition of DDK and MiDAS leads to DNA damage if those genes/pathways are not activated in euploid cells (or at least much less than in aneuploid cells)? (2) more importantly, if those mechanisms are important for both euploid and aneuploid cells, does aneuploid cells rely on them more compared with euploid cells? The differential effect shown in panel 2i is not very strong in terms of absolute change. The authors should address this concern for both DDK and MiDAS inhibition.
4. The authors report that dormant origin firing and MiDAS protect aneuploid cells from further increase in genome instability. However, this finding is based on the reversine-based short term aneuploidy model that the author use throughout the paper, as compared to established cell lines that are either diploid or aneuploid. Even though this model is of high temporal resolution, the authors do not examine whether the conclusion is limited to this specific reversine-based short term effect of aneuploidy/CIN as compared to a long term effect that would persist. In other words, is the conclusion also applicable to aneuploid cells which have grown for a while in culture? Do DDK and MiDAS prevent further DNA damage only in the short term or do they also play a role in long term aneuploid cultures? Since these questions are not addressed in the current manuscript, the authors should specify that their conclusions (as shown in the paper) only refers to a short term aneuploidy effect.
5. The authors developed a new FACS-based method to sort cycling and arrested aneuploid cells. This method will be very helpful for the aneuploidy/CIN community. One concern is: what the percentage of diploid cells in the population of "cycling aneuploid cells" given the fact that reversine likely cannot induce aneuploidy in all cells. Can the authors evaluate this issue by the single cell WGS data? Were the DNA damage level and proliferation capacity of "cycling aneuploid cells" underestimated/overestimated because of the mixed diploid cells (figure 5g-i, figure 6a-c)?
6. When the authors compared the transcriptome of cycling and arrest aneuploid cells (figure 6d-e), they report an increase in the expression of DNA repair genes in the cycling cells compared to the arrested cells. One major concern here is whether the overexpression of DNA repair genes is simply due to the fact that the cells are cycling; for example, maybe DNA repair genes (especially

those related to homologous recombination and mismatch repair) play a big role during normal DNA replication and their expression may be a consequence of cell cycle entry or replication itself. The authors can try to sort cells that are in G1 and cells that are in S/G2/M and compare their transcriptomes. Alternatively, the authors can treat the cells with CDK4/6 inhibitors to arrest the cells in G0/G1 and compare their transcriptomes to cycling cells. If a difference in expression of DNA repair genes is found, that would also be OK, but would need to be stated and discussed.

7. The title is misleading as it suggests that the authors are looking at the consequences of missegregating one chromosome (for example micronuclei formation/resolution) and the fact that they focus on the short term consequences is not clear. The authors should consider: Short-term molecular origins of genome instability following chromosome mis-segregation or Short-term molecular origins of genome instability following a chromosome mis-segregation event

Minor concerns^[1]

1. The numbers of FANCD2, RPA and pChk1 foci of control cells in figure 2 are quite different than the ones in figure 1. The authors should discuss it.
2. Why do the number of RPA foci decrease in Ctrl + DDKi group compared to Ctrl in figure 2d-e?
3. The authors should explain why DDKi + MiDASi does not induce more gammaH2AX foci compared with DDKi only in figure 4d. Also, is the effect of DDKi + MiDASi similar to the DDKi and MiDASi only (figure 4c-d).
4. Regarding the single-cell karyotype of cycling and arrested aneuploid cells (figure 5j-k): A relevant question is what is the threshold of aneuploidy to activate senescence? Can the authors plot and compare the histogram of number/frequency of gained/lost chromosome in cycling and arrested aneuploid cells to see whether there is a threshold?
5. What are the results supporting the conclusion that "it suggests that specific chromosome assortments are more prone than others to be genomically unstable, ... replication and segregation" (line 309-311).

Reviewer #2 (Remarks to the Author):

Aneuploidy, a state of karyotype imbalances, is found in more than 90% of solid cancers and in 65% of blood cancers. It arises from an increased chromosome mis-segregation rate during cell division, known as chromosomal instability (CIN). Whereas this causal link is well-established, a feedback loop in which aneuploidy can also drive CIN remains elusive. This is important to understand the origins and maintenance of genome instability in cancer, a hallmark that confers adaptive capabilities and chemoresistance, and thus holds great potential for therapeutic targeting. Here, the authors induce chromosome mis-segregation with a pulse of reversine treatment (Mps1 inhibitor) to generate aneuploid cells and follow the next cell cycles. On the first S-phase upon reversine washout, the aneuploid cells experience DNA replication stress which then impacts the quality of cell division, increasing CIN. A Dbf4-dependent kinase (DDK)-driven mechanism and mitotic DNA synthesis (MiDAS) are nevertheless used by the aneuploid cells to prevent high levels of DNA damage that would preclude them from proliferating. Yet, some daughter cells become senescent, namely those with higher karyotype complexity and DNA damage. The authors also show that cycling vs. arrested aneuploid cells display increased expression of signatures associated to DNA repair to cope with genome instability, also found in highly-proliferative cancer cells. The study brings novel insight into the aneuploidy field, it is of interest to the oncobiology area, and holds potential impact in cancer therapy. The manuscript is well-written and the methodologies combine advanced biochemical, live-cell imaging and omics experiments, adequately thought and described. There is however overinterpretation of some results that the authors should tone down, unless stronger evidence is provided to fully support the conclusions. The major concerns presented below must be addressed in order to support the publication of this manuscript.

Major comments

1. This study resorts to many sequential drug treatments aiming to synchronize cells, induce aneuploidy and interfere with cell cycle subsequent stages or processes. For each experimental

scheme shown in Figs 1-4, the authors should provide flow cytometry cell cycle profiling validating e.g.:

- the thymidine synchronization protocol
- the cell cycle dynamics upon reversine washout (S-phase and M timings)
- cell cycle dynamics under other drugs used (mimosine, aphidicolin, RO3306)

2. For the multi-FISH analysis (Fig. 1b,c) please provide the percentage of mitotic cells exhibiting gains/losses (W-CIN) and translocations (S-CIN) to better highlight the karyotype complexity already present in the 1st mitosis.

It is interesting that 10% of cells in the 1st mitosis upon Mps1i pulse have >10 errors. The authors should check senescence levels at this early stage also.

3. In page 6, lines 174-175. Unless the authors provide the % of viable cells in untreated Ctr and Aneu (without DDKi treatment), it is overinterpretation to say that aneuploid cells are more sensitive to DDK inhibition. The fold change in the % of viable cells in Ctr and Aneu conditions following DDKi treatment could be the same.

4. The non-random distribution of DNA damage between aneuploid daughter cells might well account for different proliferation rates. But unless daughters are tracked for their fate, it is speculative whether this mechanism primarily drives senescence. Micronucleation might also account for different daughter cell fates. Interestingly, MN appear specifically induced in Aneu vs. RS conditions (Fig. 3I) and prevented by MiDASi.

5. Please avoid the use of 'novel' when referring to SA-b-Gal cell sorting.

What is the percentage of senescent cells immediately after DMSO/reversine treatment and along the subsequent cell cycles? Is there an accrual of senescent cells with the increasing genomic instability along the cycles?

6. In page 9, line 292. The experiments do not demonstrate that asymmetric inheritance of DNA damage contributes to cell cycle arrest in aneuploid cells. Symmetric divisions generating daughter cells both with damage/karyotype complexity above a certain threshold could also lead to senescence. This introductory sentence should be removed from the section.

7. The last section of Results raises major concerns. Cycling aneuploid cells are fitter than arrested aneuploid cells, thus is not surprising that p53 and inflammation gene expression are lower, and cell cycle and DNA repair gene expression higher. Control cells must be included in the RNAseq analysis for comparison. If control cells exhibit higher cell cycle and DNA repair gene expression, as well as faster decay kinetics of 53BP1/gH2AX in comparison to cycling aneuploid cells, an activation of DNA damage repair pathways is unlikely.

Moreover, still in this section, for the CCLE database analysis correlating doubling time with DNA repair gene overexpression, an association with the aneuploidy index is missing, i.e., which are expected to double faster and overexpress DNA repair genes – the highly or the low aneuploid cell lines?

Anyway, the immediate consequences of aneuploidy disclosed in this study could be different from long-term adapted aneuploid cancer cells.

Minor comments

1. It is unclear whether thymidine block was done for multi-FISH karyotyping (Fig. 1b) (check different description in Results section page 4 and in Methods section page 13).

2. I suggest to remove the word 'single' from the title.

3. In Fig. 1g-l, as well as in Fig. 2b-g, the percentage of S-phase cells exhibiting a number of FANCD2, RPA, p-Chk1 foci above the control threshold (mean value) should be also graphically represented. Same applies to Fig. 3f, Fig.3i, Fig. 4c,d and Fig. 5h,i.

4. In page 5, line 163. Please refer how the G1 arrest was induced (I believe mimosine was used).

Note that minosine is reported to induce DNA breaks (Mikhailov I et al. *Mutat Res* 2000 459:299-306), which might explain the increased number of FANCD2, RPA, p-Chk1 foci observed in Fig.2b-g vs. Fig. 1g-l.

Reviewer #3 (Remarks to the Author):

In this manuscript, De Feudis et al induce aneuploidy in RPE cells using Mps1 inhibition following cell cycle synchronization. They report increased DNA replication stress in aneuploid cells that enhances their reliance on DDK and MiDAS. Separation of senescent aneuploid cells based on β -galactosidase activity permitted them to conclude that senescent cells have increased amounts of DNA damage and karyotype complexity. Cycling aneuploid RPE cells, as well as rapidly cycling cancer cell lines, have higher levels of DNA damage repair proteins. They conclude that aneuploidy induces genome instability due to DNA replication stress and that both DNA damage and karyotype complexity contribute to aneuploidy-induced senescence.

The interplay between aneuploidy and CIN is interesting. CIN results in aneuploidy, though aneuploid cells may be eliminated through selection. Whether and how aneuploidy results in ongoing CIN is less well-characterized, but Dr. Santaguida and others have previously shown convincingly that aneuploidy results in DNA damage, prolonged S phase, reduced DNA replication fork rate, increased fork stalling, increased origin firing, increased reliance on DDK, insufficiency of DNA replication factors, and entry into mitosis in the presence of DNA damage in yeast and human cells (Ohashi et al, *Nat Commun* 2015; Blank et al, *Mol Biol Cell* 2015; Torres et al, *Science* 2007; Burrell et al, *Nature* 2013; Passerini et al, *Nat Commun* 2016; Lamm et al, *Cell Stem Cell* 2016; Santaguida et al, *Dev Cell* 2017). The method to separate senescent cells presented here is novel. But the extent of the previous evidence reporting very similar findings limits the impact of the current manuscript.

Major concerns

- 1) There is an apparent lack of experimental rigor. For example,
 - a. All of the conclusions about the impact of aneuploidy are made after cell cycle synchronization and Mps1 inhibition. Given that cell cycle synchronization induces DNA damage (Wong and Stearns, *BMC Cell Biology* 2005) and Mps1 inhibition reduces DNA damage repair (Yu et al, *Nucleic Acids Research* 2016; Maachani et al, *Mol Cancer Res* 2015; Wei et al, *J Biol Chem* 2005), it seems necessary for an experimentally rigorous approach to demonstrate that the rate and extent of DNA damage and repair is not affected by the methodology.
 - b. Sample sizes, including the number of cells analyzed per biological replicate, are largely not described (eg Fig. 1c, h, j, l, n, o, 2c, e, g, i, 3c, d, I, k, l, 4c, d, I, g, j, 5f, h, i, 6c, f, g). Where described, the data appear to have come from a single biological replicate (eg Fig. 1f "108 and 95 forks were analyzed in the control and in the aneuploid sample, respectively"). For dot plots such as 1h, it would be helpful to show data from separate biological replicates in different colors ("superplots" Goedhart, *Mol Biol Cell* 2021) or at least to show the averages/medians of each of the biological replicates in addition to the individual data points. Statistics appear to have been performed from each cell as opposed to each biological replicate.
 - c. Though the data in Fig. 5 show that senescence correlates with DNA damage and karyotype complexity, this correlation does not show causation and is not sufficient for the conclusion that "cell cycle arrest in aneuploid cells is due to both DNA damage and karyotype complexity" or that "cell cycle arrest in aneuploid cells is due to not only the degree of karyotype aberrations but also to the levels of DNA damage harbored by the cells".
 - d. The conclusions about MiDAS don't appear to be supported by the data in Fig. 4c-d. Lines 238-240 say that "the number of both FANCK2 and gH2AX foci was significantly higher in aneuploid cells in which DDK and MiDAS were inhibited compared to aneuploid cells in which only either DDK or MiDAS was hindered (Fig. 4b-d)." But Fig. 4d indicates that in aneuploid cells there is no difference in gH2AX foci between DDKi and the combination of DDKi with MiDASi. Line 240 says "inhibition of those pathways led to an increase in FANCD2 and gH2AX foci also in euploid cells", but MiDASi doesn't increase FANCD2 or gH2AX in control cells (Fig. 4c-d).
 - e. It doesn't seem appropriate to compare FANCD2 or RPA foci in nuclei that are in very different

stages of S phase, as in Fig. 1g-l.

f. Fig. ED1b appears to present a biological replicate of the data presented in Fig. 1c, but results are rather different. Are these really biological replicates or is there a difference here that is not readily apparent?

2. Fig. 6a-c nicely shows that aneuploid cycling cells don't show an increase in mitotic errors, and thus that aneuploidy does not invariably lead to CIN, which is summarized in the model in Fig. 7 that says that aneuploid cells with induced genome instability arrest, while those that cycle remain genomically stable. However, this seems to contradict the title and suggest that the genomically unstable cells will not persist/divide to generate ongoing instability in the population.

Minor comments

The schematics of the experimental design are helpful, but are often missing details about the cell cycle synchronization and the amount of time between different steps. Please include these.

Why is the rate of translocations so high in control aneuploid cells (~30%, Fig. 4g)? Simply inhibiting the SAC for 1 mitosis would not be expected to cause this. Perhaps the cells were analyzed after several days and this is a result of subsequent damage, but the timing with which this experiment was done is not apparent.

The word "length" is misspelled multiple times in Fig. 3a and in Fig. 3c.

Reviewer #4 (Remarks to the Author):

The manuscript by Santaguida and colleagues describes the strategies used by aneuploid cells to protect themselves from excessive replication stress. The authors provide evidence that a combination of DKK-dependent origin firing and MiDAS-driven protection against under-replication suppress aneuploidy-induced chromosome instability. The findings are interesting and relevant to our understanding of aneuploidy in cancer.

The MiDAS analysis using individual metaphase chromosomes showed an atypical and unusual pattern of EdU foci. Most foci seem to be individual and located apparently at or very close to the ends of the chromosomes. It would be informative to identify whether the major loci affected by the aneuploidy are telomeric or located at defined locations on specific chromosomes. This could be achieved by combining EdU analysis with either immunofluorescence for TRF2 or PNA FISH for telomeric repeat DNA.

REVIEWER COMMENTS

Reviewer #1 (Remarks to the Author):

The manuscript by De Feudis et al., provides interesting novel insights into the mechanisms of short-term genome instability following induction of chromosome mis-segregation and aneuploidy. By focusing on the 1st and second cycle after missegregation, the authors offer a high temporal resolution on the underlying mechanism of how aneuploidy may induce genomic instability. For publication, the paper would benefit from additional experiments and edits on the text to strengthen the conclusions and discuss the remaining open questions (i.e. what the study does not address, such as the long term consequences of CIN/aneuploidy).

We thank Reviewer #1 for her/his insightful and constructive suggestions. We appreciate that she/he acknowledged the “interesting novel insights into the mechanisms of short-term genome instability following induction of chromosome mis-segregation and aneuploidy” and our efforts to “offer a high temporal resolution on the underlying mechanism of how aneuploidy may induce genomic instability”. We are grateful to her/him for suggesting experiments and edits to the text aimed at improving the manuscript. We have followed those suggestions, as outlined below.

Major concerns

1. A general concern of several results shown in the paper is the fact that many analyses rely on using drugs/chemicals to inhibit genes or pathways and in most cases are not validated by genetic approaches (based on CRISPRi or cDNA overexpression). While the time sensitive experiments based on reversine may be tricky to recapitulate genetically, they should be repeated and validated using alternative MPS1 inhibitors. For the other experiments, including inhibitors of DDK and MiDAs, the key experiments (especially Fig. 2 and 3) should be confirmed using a genetic approach.

We thank the reviewer for raising this valid point. We agree and followed the experimental suggestions. In particular:

1. We employed AZ3146 as an alternative Mps1 inhibitor, as shown in Extended Data Fig.3a, c-h, Extended Data Fig.5a-g and Extended Data Fig.6a-d. Importantly, these experiments validated our previous findings.
2. We utilized an analogue sensitive allele of Cdc7 to interfere with the function of DDK, as shown in Extended Data Fig. 3a-h. These experiments confirmed our results obtained using a chemical inhibitor of DDK.
3. We inhibited MiDAS by depleting the non-catalytic subunit of DNA polymerase delta POLD3, which has been shown to be essential for the two MiDAS pathways described so far (Minocherhomji et al Nature 2015; Garribba et al PNAS 2020). These data are now shown in Extended Data Fig. 5a,g. These experiments validated our previous results.

2. Examining the short-term effects of aneuploidy on genome integrity is an important question in the field. However, it is also very difficult to distinguish the effect of CIN and aneuploidy. In figure 1, the authors induce aneuploidy by pulsing the cells with reversine which induces the missegregation of chromosomes and then compare the cells from the 1st mitosis and 2nd mitosis. The problem is the following: are the observed karyotypes/cellular changes (or other difference) observed in 2nd mitosis the consequence of aneuploidy in the 1st mitosis or of long-term disruption of chromosome stability by reversine? In other words, can reversine impair chromosome stability in a long-term and aneuploidy-

independent manner even though it had been washed out? The authors should discuss this potential caveat.

We thank the Reviewer for pointing this out. We have discussed this potential issue in the text (lines 123-125).

3. As the authors report in figure 4d and related panels, inhibition of DDK and MiDAS can lead to an increase in FANCD2 and gamma-H2AX foci in euploid cells. This means those two phenotypes are not specific to aneuploid cells. This raises two questions: (1) why does the inhibition of DDK and MiDAS leads to DNA damage if those genes/pathways are not activated in euploid cells (or at least much less than in aneuploid cells)? (2) more importantly, if those mechanisms are important for both euploid and aneuploid cells, does aneuploid cells rely on them more compared with euploid cells? The differential effect shown in panel 2i is not very strong in terms of absolute change. The authors should address this concern for both DDK and MiDAS inhibition.

We thank the reviewer for this comment. We think that even euploid cells might experience some degree of DNA damage during S phase (as expected during replication of DNA) and might rely on DDK and MiDAS, although at much lower extent compared to aneuploid cells. For this reason, inhibition of DDK and MiDAS can lead to a slight increase in FANCD2 and gamma-H2AX foci (although only inhibition of DDK leads to a difference that is statistically significant). We are also discussing this in the text (lines 268-270). Interestingly, inhibition of DDK through a different approach - namely, a Cdc7 analogue sensitive allele - also led to an increase in FANCD2 foci in euploid cells (Extended Data Figure 3c, d).

We agree with the Reviewer that the effect shown in Figure 2i is not very strong but we found it to be reproducible and the difference to be statistically significant.

4. The authors report that dormant origin firing and MiDAS protect aneuploid cells from further increase in genome instability. However, this finding is based on the reversine-based short term aneuploidy model that the author use throughout the paper, as compared to established cell lines that are either diploid or aneuploid. Even though this model is of high temporal resolution, the authors do not examine whether the conclusion is limited to this specific reversine-based short term effect of aneuploidy/CIN as compared to a long term effect that would persist. In other words, is the conclusion also applicable to aneuploid cells which have grown for a while in culture? Do DDK and MiDAS prevent further DNA damage only in the short term or do they also play a role in long term aneuploid cultures? Since these questions are not addressed in the current manuscript, the authors should specify that their conclusions (as shown in the paper) only refers to a short term aneuploidy effect.

We completely agree with this suggestion and thank the Reviewer for this comment. We are making clear that our results refer to the immediate effects of aneuploidy by changing the title, which now contains "short-term", and in lines 37, 409 and 443.

5. The authors developed a new FACS-based method to sort cycling and arrested aneuploid cells. This method will be very helpful for the aneuploidy/CIN community. One concern is: what the percentage of diploid cells in the population of "cycling aneuploid cells" given the fact that reversine likely cannot induce aneuploidy in all cells. Can the authors evaluate this issue by the single cell WGS data? Were the DNA damage level and proliferation capacity of "cycling aneuploid cells" underestimated/overestimated because of the mixed diploid cells (figure 5g-i, figure 6a-c)?

We thank the Reviewer for recognizing our method to sort cycling and arrested aneuploid cells as “very helpful for the aneuploidy/CIN community”. Following her/his suggestion we checked the % of diploids in the two aneuploid populations and found it to be 24% (12 out of 50) in cycling aneuploid cells and 13% (7 out of 53) in arrested aneuploid cells. We are mentioning this in lines 375-376. To make sure that this fraction of diploid cells in the aneuploid populations did not affect the interpretation of the results, we decided to run a comparison of the transcriptome of cycling aneuploid cells vs. diploid cells, as also asked in point #6 of this Reviewer comments and in point #7 of Reviewer #2. As outlined below (point #6) and in the text (lines 380-381), cycling aneuploid cells showed higher expression of DNA repair genes, even when compared to euploid cells. Thus, we conclude that the fraction of diploid cells present in both aneuploid populations might play only a minor role, if anything, in the described phenotypes.

6. When the authors compared the transcriptome of cycling and arrest aneuploid cells (figure 6d-e), they report an increase in the expression of DNA repair genes in the cycling cells compared to the arrested cells. One major concern here is whether the overexpression of DNA repair genes is simply due to the fact that the cells are cycling; for example, maybe DNA repair genes (especially those related to homologous recombination and mismatch repair) play a big role during normal DNA replication and their expression may be a consequence of cell cycle entry or replication itself. The authors can try to sort cells that are in G1 and cells that are in S/G2/M and compare their transcriptomes. Alternatively, the authors can treat the cells with CDK4/6 inhibitors to arrest the cells in G0/G1 and compare their transcriptomes to cycling cells. If a difference in expression of DNA repair genes is found, that would also be OK, but would need to be stated and discussed.

We thank the Reviewer for pointing this out. To address this issue we decided to compare cycling aneuploid cells to euploid ones. Importantly, even when performing this comparison, cycling aneuploid cells showed higher expression of DNA repair genes. We are discussing this in the text (lines 380-381) and showing it in Extended Data Fig. 8g, h.

7. The title is misleading as it suggests that the authors are looking at the consequences of missegregating one chromosome (for example micronuclei formation/resolution) and the fact that they focus on the short term consequences is not clear. The authors should consider: Short-term molecular origins of genome instability following chromosome mis-segregation or Short-term molecular origins of genome instability following a chromosome mis-segregation event.

We appreciate this comment and have changed the title to “Short-term molecular consequences of chromosome mis-segregation for genome stability”.

Minor concerns

1. The numbers of FANCD2, RPA and pChk1 foci of control cells in figure 2 are quite different than the ones in figure 1. The authors should discuss it.

These experiments have been performed over the course of a multi-year project utilizing different frozen vials of the same cell line (we keep cell lines in culture for no more than 5/6 weeks). Thus, it is entirely possible that these differences reflect biological variations intrinsic to cell culture methods.

2. Why do the number of RPA foci decrease in Ctrl + DDKi group compared to Ctrl in figure 2d-e?

This is in agreement with previous findings (Sasi et al., Neoplasia 2018; Jones et al., Mol Cell 2021) showing that DDKi blocks RPA recruitment (a trend also maintained by using a Cdc7 analogue sensitive allele, although it did not reach statistical significance, Extended Data Figure 3e, f). We are mentioning this in the text (lines 188-191).

3. The authors should explain why DDKi + MiDASi does not induce more gammaH2AX foci compared with DDKi only in figure 4d. Also, is the effect of DDKi + MiDASi similar to the DDKi and MiDASi only (figure 4c-d).

The data in Figure 4d were showing a trend of more gammaH2AX in DDKi + MiDASi compared to DDKi only, although the difference was not statistically significant. Thus, we decided to expand our dataset analyzing more cells and found that the effect of DDKi + MiDASi is significantly higher than DDKi alone. Further, our data also show that the effect of DDKi + MiDASi is significantly higher than DDKi or MiDASi alone, in all tested conditions.

4. Regarding the single-cell karyotype of cycling and arrested aneuploid cells (figure 5j-k): A relevant question is what is the threshold of aneuploidy to activates senescence? Can the authors plot and compare the histogram of number/frequency of gained/lost chromosome in cycling and arrested aneuploid cells to see whether there is a threshold?

We tried to do so but we did not find a significant trend of particular gained/lost chromosomes. We are showing these data in Extended Data Figure 7g, h and mentioning in the text (lines 347-348).

5. What are the results supporting the conclusion that “it suggests that specific chromosome assortments are more prone than others to be genomically unstable, ... replication and segregation” (line 309-311).

This was speculative. Thus, we removed the sentence.

Reviewer #2 (Remarks to the Author):

Aneuploidy, a state of karyotype imbalances, is found in more than 90% of solid cancers and in 65% of blood cancers. It arises from an increased chromosome mis-segregation rate during cell division, known as chromosomal instability (CIN). Whereas this causal link is well-established, a feedback loop in which aneuploidy can also drive CIN remains elusive. This is important to understand the origins and maintenance of genome instability in cancer, a hallmark that confers adaptive capabilities and chemoresistance, and thus holds great potential for therapeutic targeting.

Here, the authors induce chromosome mis-segregation with a pulse of reversine treatment (Mps1 inhibitor) to generate aneuploid cells and follow the next cell cycles. On the first S-phase upon reversine washout, the aneuploid cells experience DNA replication stress which then impacts the quality of cell division, increasing CIN. A Dbf4-dependent kinase (DDK)-driven mechanism and mitotic DNA synthesis (MiDAS) are nevertheless used by the aneuploid cells to prevent high levels of DNA damage that would preclude them for proliferating. Yet, some daughter cells become senescent, namely those with higher karyotype complexity and DNA damage. The authors also show that cycling vs. arrested aneuploid cells display increased expression of signatures associated to DNA repair to cope with genome instability, also found in highly-proliferative cancer cells.

The study brings novel insight into the aneuploidy field, it is of interest to the oncobiology area, and holds potential impact in cancer therapy. The manuscript is well-written and the methodologies combine advanced biochemical, live-cell imaging and omics experiments, adequately thought and described. There is however overinterpretation of some results that the authors should tone down, unless stronger

evidence is provided to fully support the conclusions. The major concerns presented below must be addressed in order to support the publication of this manuscript.

We thank Reviewer#2 for recognizing that our study “brings novel insight into the aneuploidy field, it is of interest to the oncobiology area, and holds potential impact in cancer therapy”. We also appreciate that she/he found the manuscript to be “well-written and the methodologies combine advanced biochemical, live-cell imaging and omics experiments, adequately thought and described”. We also thank her/him for providing valuable suggestions to improve the manuscript that we followed, as outlined below.

Major comments

1. This study resorts to many sequential drug treatments aiming to synchronize cells, induce aneuploidy and interfere with cell cycle subsequent stages or processes. For each experimental scheme shown in Figs 1-4, the authors should provide flow cytometry cell cycle profiling validating e.g.:

- the thymidine synchronization protocol
- the cell cycle dynamics upon reversine washout (S-phase and M timings)
- cell cycle dynamics under other drugs used (mimosine, aphidicolin, RO3306)

We thank the Reviewer for pointing at this. We have followed her/his suggestions and, in particular, have included cell cycle profiles showing:

1. Thymidine synchronization and wash-out protocol in Extended Data Figure 1a, b;
2. Cell cycle dynamics under mimosine in Extended Data Figure 2a, b;
3. Cell cycle dynamics under aphidicolin in Extended Data Figure 2f;
4. Cell cycle dynamics under RO3306 in Extended Data Figure 4c, d.

We also note that cell cycle dynamics upon reversine wash-out is already present in Figure 3a-d. These panels show a live-cell imaging experiment in which S-phase length (Figure 3c) and M timing (Figure 3d) were calculated using a hTERT RPE-1 cell line stably expressing PCNA-GFP and H2b-RFP.

2. For the multi-FISH analysis (Fig. 1b,c) please provide the percentage of mitotic cells exhibiting gains/losses (W-CIN) and translocations (S-CIN) to better highlight the karyotype complexity already present in the 1st mitosis.

It is interesting that 10% of cells in the 1st mitosis upon Mps1i pulse have >10 errors. The authors should check senescence levels at this early stage also.

We thank the Reviewer for suggesting the analysis of the percentage of mitotic cells exhibiting gains/losses (W-CIN) and translocations (S-CIN). We have included this in Figure 1d. Interestingly, this analysis revealed that numerical aneuploidies accounted for most of the measured aneuploidy events, both in the 1st and in the 2nd mitosis. We are also making this comment in the text (lines 126-127).

As suggested, we also checked senescence levels after the 1st mitosis and did not find an increase (Extended Data Figure 7a).

3. In page 6, lines 174-175. Unless the authors provide the % of viable cells in untreated Ctr and Aneu (without DDKi treatment), it is overinterpretation to say that aneuploid cells are more sensitive to DDK inhibition. The fold change in the % of viable cells in Ctr and Aneu conditions following DDKi treatment could be the same.

We have evaluated the % of viable cells in untreated Ctr and Aneu (without DDKi treatment) and found no major differences between the two samples (about 95% of viable cells in both cases). We are showing these data in Extended Data Figure 3i and mentioning it in the text (lines 194-195).

4. The non-random distribution of DNA damage between aneuploid daughter cells might well account for different proliferation rates. But unless daughters are tracked for their fate, it is speculative whether this mechanism primarily drives senescence. Micronucleation might also account for different daughter cell fates. Interestingly, MN appear specifically induced in Aneu vs. RS conditions (Fig. 3l) and prevented by MiDASi.

We thank the Reviewer for this comment that prompted us to follow the fate of aneuploid daughter cells by live cell imaging. For this, we employed a hTERT RPE-1 cell line expressing H2b-GFP and RNF168-miRFP – as a marker of DNA damage - and assessed DNA damage inheritance in aneuploid cells over a 5-day timeframe. By doing this, we found RNF168 to be asymmetrically partitioned in more than 20% of daughter cells, in agreement with our findings obtained by monitoring FANCD2 and γ H2AX in fixed cells. Also, DNA damage transmission from mother to daughter cells was more likely to happen in aneuploid cells eventually displaying impaired proliferative capacity and the incidence of cell divisions in which DNA damage was asymmetrically distributed between daughter cells was higher in cells with low proliferative capacity. These data are shown in Extended Data Fig. 6a-f and discussed in the text (lines 287-298). We do agree that it is also possible that micronucleation (together with other aneuploidy-associated features) might play a role in this process and are discussing this possibility in the text (lines 298-301).

5. Please avoid the use of 'novel' when referring to SA-b-Gal cell sorting.

What is the percentage of senescent cells immediately after DMSO/reversine treatment and along the subsequent cell cycles? Is there an accrual of senescent cells with the increasing genomic instability along the cycles?

We followed Reviewer's suggestion and removed the word 'novel' when referring to SA-b-Gal cell sorting. We have also measured senescence levels after DMSO/reversine treatment and along the subsequent cell cycles. These data are now included in Extended Data Figure 7a and show that there is an increase in SA-b-Gal 72 hours after the induction of chromosome mis-segregation. We are also commenting on this in the text (314-317).

6. In page 9, line 292. The experiments do not demonstrate that asymmetric inheritance of DNA damage contributes to cell cycle arrest in aneuploid cells. Symmetric divisions generating daughter cells both with damage/karyotype complexity above a certain threshold could also lead to senescence. This introductory sentence should be removed from the section.

We agree with the Reviewer and have removed the mentioned sentence.

7. The last section of Results raises major concerns. Cycling aneuploid cells are fitter than arrested aneuploid cells, thus is not surprising that p53 and inflammation gene expression are lower, and cell cycle and DNA repair gene expression higher. Control cells must be included in the RNAseq analysis for comparison. If control cells exhibit higher cell cycle and DNA repair gene expression, as well as faster decay kinetics of 53BP1/ γ H2AX in comparison to cycling aneuploid cells, an activation of DNA damage repair pathways is unlikely.

We thank the Reviewer for pointing this out. As suggested by her/him, we compared cycling aneuploid cells to euploid ones. Importantly, even when performing this comparison, cycling aneuploid cells showed higher expression of DNA repair genes. We are discussing this in the text (lines 373-381) and showing it in Extended Data Fig. 8g, h.

Further, we also found a faster decay kinetics of γ H2AX levels - but not 53BP1 - in the aneuploid cycling population compared to euploid controls. We have included these results in Extended Data Fig. 8c,d and commented in the text (368-371).

Moreover, still in this section, for the CCLE database analysis correlating doubling time with DNA repair gene overexpression, an association with the aneuploidy index is missing, i.e., which are expected to double faster and overexpress DNA repair genes – the highly or the low aneuploid cell lines?

We thank the Reviewer for this comment. We have looked into this and want to highlight that, on average, highly-aneuploid cancer cells have lower proliferative capacity compared to near-euploid ones. This has been recently reported (Replogle et al., PNAS 2020) and the relevant graph from that publication is pasted below.

[redacted]

Panel and figure legend from Replogle et al., PNAS 2020.

Figure legend: (A) Histogram of doubling times of highly aneuploid (top quartile) and near-euploid cell lines (bottom quartile) from the Cancer Cell Line Encyclopedia obtained from the Broad Institute's RNAi dependency screen. P value represents a two-sided Wilcoxon rank-sum test.

We also tried to correlate the expression of DNA repair genes with ploidy status and did not find significant differences between the DNA damage repair signatures of the highly-aneuploid cell lines compared to near-euploid ones.

Anyway, the immediate consequences of aneuploidy disclosed in this study could be different from long-term adapted aneuploid cancer cells.

We completely agree with this comment. We are making this clear by changing the title which now contains “short-term” and in lines 37, 409 and 443.

Minor comments

1. It is unclear whether thymidine block was done for multi-FISH karyotyping (Fig. 1b) (check different description in Results section page 4 and in Methods section page 13).

We thank the Reviewer for pointing this out. There was a mistake in the Methods, which has now been corrected (lines 474-478).

2. I suggest to remove the word 'single' from the title.

Agreed. We removed the word 'single' from the title.

3. In Fig. 1g-l, as well as in Fig. 2b-g, the percentage of S-phase cells exhibiting a number of FANCD2, RPA, p-Chk1 foci above the control threshold (mean value) should be also graphically represented. Same applies to Fig. 3f, Fig.3i, Fig. 4c,d and Fig. 5h,i.

We have tried multiple ways of displaying our results, including the graphical representation suggested by the Reviewer. We found the superplots suggested by Reviewer #3 in her/his comment #1b to offer a comprehensive view of the data and decided to keep this representation of the data.

4. In page 5, line 163. Please refer how the G1 arrest was induced (I believe mimosine was used). Note that mimosine is reported to induce DNA breaks (Mikhailov I et al. Mutat Res 2000 459:299-306), which might explain the increased number of FANCD2, RPA, p-Chk1 foci observed in Fig.2b-g vs. Fig. 1g-l.

We thank the Reviewer for pointing at this. We want to clarify that in both cases (Fig. 1g-l and Fig.2b-g) mimosine was employed, as also outlined in the text and in the experimental schemes in Fig. 1e and Fig. 2a. Following Reviewer remarks on mimosine-dependent induction of DNA breaks, we carefully looked into this and found that mimosine leads to accumulation of mild levels of DNA damage, as pointed by the Reviewer. However, this damage was efficiently repaired with the same efficiency in reversine or vehicle control treated cells. We are showing these results in Extended Data Fig. 2c-e and commented in the text (131-133).

Reviewer #3 (Remarks to the Author):

In this manuscript, De Feudis et al induce aneuploidy in RPE cells using Mps1 inhibition following cell cycle synchronization. They report increased DNA replication stress in aneuploid cells that enhances their reliance on DDK and MiDAS. Separation of senescent aneuploid cells based on b-galactosidase activity permitted them to conclude that senescent cells have increased amounts of DNA damage and karyotype complexity. Cycling aneuploid RPE cells, as well as rapidly cycling cancer cell lines, have higher levels of DNA damage repair proteins. They conclude that aneuploidy induces genome instability due to DNA replication stress and that both DNA damage and karyotype complexity contribute to aneuploidy-induced senescence.

The interplay between aneuploidy and CIN is interesting. CIN results in aneuploidy, though aneuploid cells may be eliminated through selection. Whether and how aneuploidy results in ongoing CIN is less well-characterized, but Dr. Santaguida and others have previously shown convincingly that aneuploidy results in DNA damage, prolonged S phase, reduced DNA replication fork rate, increased fork stalling, increased origin firing, increased reliance on DDK, insufficiency of DNA replication factors, and entry into mitosis in the presence of DNA damage in yeast and human cells (Ohashi et al, Nat Commun 2015; Blank et al, Mol Biol Cell 2015; Torres et al, Science 2007; Burrell et al, Nature 2013; Passerini et al, Nat Commun 2016; Lamm et al, Cell Stem Cell 2016; Santaguida et al, Dev Cell 2017). The method to separate senescent cells presented here is novel. But the extent of the previous evidence reporting very similar findings limits the impact of the current manuscript.

We thank Reviewer#3 for her/his insightful suggestions that we followed, as outlined below. S/he also was concerned by the fact that “the extent of the previous evidence reporting very similar findings limits the impact of the current manuscript”. We would like to mention that the other Reviewers found our work novel and insightful. In particular, they claim that:

- It provides “interesting novel insights into the mechanisms of short-term genome instability following induction of chromosome mis-segregation and aneuploidy” (Reviewer#1).
- It “brings novel insight into the aneuploidy field, it is of interest to the oncobiology area, and holds potential impact in cancer therapy” (Reviewer#2).
- “The findings are interesting and relevant to our understanding of aneuploidy in cancer” (Reviewer#4).

We have made several changes to the manuscript and added data that have strengthened our conclusions, also thanks to Reviewer#3 comments. Thus, we hope s/he now recognizes the importance of our findings.

Major concerns

1) There is an apparent lack of experimental rigor. For example,

a. All of the conclusions about the impact of aneuploidy are made after cell cycle synchronization and Mps1 inhibition. Given that cell cycle synchronization induces DNA damage (Wong and Stearns, BMC Cell Biology 2005) and Mps1 inhibition reduces DNA damage repair (Yu et al, Nucleic Acids Research 2016; Maachani et al, Mol Cancer Res 2015; Wei et al, J Biol Chem 2005), it seems necessary for an experimentally rigorous approach to demonstrate that the rate and extent of DNA damage and repair is not affected by the methodology.

We thank the Reviewer for pointing this out and for suggesting to demonstrate that the rate and extent of DNA damage and repair is not affected by cell cycle synchronization and Mps1 inhibition. We have followed these experimental suggestions and evaluated the consequences of:

1. Thymidine block and release in presence of reversine (or vehicle control) using γ H2AX as a marker of DNA damage. As expected, we found that thymidine block induced mild levels of DNA damage and, importantly, this was fully repaired 12 hours after washout with a similar kinetics in cells exposed to Mps1 inhibitor or vehicle control. These results are shown in Extended Data Fig. 1c-e and discussed in the text (lines 107-113).
2. Mimosine block and release in cells pulsed with Mps1i or vehicle control. We used γ H2AX as a marker of DNA damage and found that mimosine leads to accumulation of mild levels of DNA damage. However, this damage was efficiently repaired with the same efficiency in reversine or vehicle control treated cells. We are showing these results in Extended Data Fig. 2c-e and commented in the text (131-133).
3. RO3306 block and did not find any significant increase in the levels of DNA damage. These results are shown in Extended Data Fig. 4e-g and discussed in the text (lines 229-230).

b. Sample sizes, including the number of cells analyzed per biological replicate, are largely not described (eg Fig. 1c, h, j, l, n, o, 2c, e, g, i, 3c, d, l, k, l, 4c, d, l, g, j, 5f, h, i, 6c, f, g). Where described, the data appear to have come from a single biological replicate (eg Fig. 1f “108 and 95 forks were analyzed in the control and in the aneuploid sample, respectively”). For dot plots such as 1h, it would be helpful to show data from separate biological replicates in different colors (“superplots” Goedhart, Mol Biol Cell 2021) or at least to show the averages/medians of each of the biological replicates in addition to the individual data points. Statistics appear to have been performed from each cell as opposed to each biological replicate.

We thank the Reviewer for pointing at this. In figure legends, we now provide a clear description of the sample size, as well as the number of biological replicates and the statistical test performed for each panel. We note that the vast majority of experiments were conducted in three biological replicates (with a few exceptions clearly stated in figure legends). Regarding the experiment mentioned by the Reviewer (now Fig. 1g) we note that EM analysis of replication forks is obtained through a lengthy experimental procedure and analysis and is meant as a qualitative integration of the results of this work through an orthogonal approach. Were this experiment alone considered crucial to sustain the conclusions of this paper, we would surely invest our time and resources in performing multiple biological replicates, which however, would require for us a longer time than what's normally allocated to address the reviewers' concerns.

We also thank the Reviewer for the very helpful suggestion regarding the dot plots, which we followed. We have now included the average for each biological replicate on the graphs using a different color.

c. Though the data in Fig. 5 show that senescence correlates with DNA damage and karyotype complexity, this correlation does not show causation and is not sufficient for the conclusion that "cell cycle arrest in aneuploid cells is due to both DNA damage and karyotype complexity" or that "cell cycle arrest in aneuploid cells is due to not only the degree of karyotype aberrations but also to the levels of DNA damage harbored by the cells".

We thank the Reviewer for this comment and agree with her/his concern. Thus, we tuned down these claims by changing "cell cycle arrest in aneuploid cells is due to both DNA damage and karyotype complexity" to "DNA damage and karyotype complexity might be responsible, at least partially, for cell cycle defects of aneuploid cells" (lines 338-339). Further, we changed "cell cycle arrest in aneuploid cells is due to not only the degree of karyotype aberrations but also to the levels of DNA damage harbored by the cells" to "increasing degrees of karyotype aberrations together with high levels of DNA could contribute, at least partially, to cell cycle arrest of aneuploid cells" (lines 398-399).

d. The conclusions about MiDAS don't appear to be supported by the data in Fig. 4c-d. Lines 238-240 say that "the number of both FANCK2 and gH2AX foci was significantly higher in aneuploid cells in which DDK and MiDAS were inhibited compared to aneuploid cells in which only either DDK or MiDAS was hindered (Fig. 4b-d)." But Fig. 4d indicates that in aneuploid cells there is no difference in gH2AX foci between DDKi and the combination of DDKi with MiDASi.

We thank the Reviewer for highlighting this. The data in Figure 4d were showing a trend of more gammaH2AX in DDKi + MiDASi compared to DDKi only, although the difference was not statistically significant. Thus, we decided to expand our dataset analyzing more cells and found that the effect of DDKi + MiDASi is significantly higher than DDKi alone.

Further, our data also show that the effect of DDKi + MiDASi is significantly higher than DDKi or MiDASi alone, in all tested conditions.

Line 240 says "inhibition of those pathways led to an increase in FANCD2 and gH2AX foci also in euploid cells", but MiDASi doesn't increase FANCD2 or gH2AX in control cells (Fig. 4c-d).

Thanks for pointing at this. We meant that inhibition of both pathways led to an increase in FANCD2 and gH2AX foci also in euploid cells. We changed the text accordingly (line 269).

e. It doesn't seem appropriate to compare FANCD2 or RPA foci in nuclei that are in very different stages of S phase, as in Fig. 1g-l.

We thank the Reviewer for this comment. Cells were chosen based on their positiveness for EdU staining, with the goal to select S phase cells on which number of foci were counted. This allowed us to perform a completely unbiased selection of cells (with respect to DNA damage markers), since the only requirement to be selected was their positive signal for EdU. We also would like to highlight that EdU was present in the medium for just 30 minutes, so we labeled only cells that were in S phase (after cell cycle synchronization) in that short time window. Finally, as pointed out by the Reviewer, it is possible that there were differences in the stages of S phase in which cells were but given the high number of cells analyzed from different biological replicates, we think it is unlikely that this affected the interpretation of the results.

f. Fig. ED1b appears to present a biological replicate of the data presented in Fig. 1c, but results are rather different. Are these really biological replicates or is there a difference here that is not readily apparent?

Extended Data Figure 1b, which now is Extended Data Figure 1i, shows the results of a DMSO pulse, whereas Figure 1c refers to Mps1i. This is stated in figure legend.

2. Fig. 6a-c nicely shows that aneuploid cycling cells don't show an increase in mitotic errors, and thus that aneuploidy does not invariably lead to CIN, which is summarized in the model in Fig. 7 that says that aneuploid cells with induced genome instability arrest, while those that cycle remain genomically stable. However, this seems to contradict the title and suggest that the genomically unstable cells will not persist/divide to generate ongoing instability in the population.

We thank the Reviewer for this very helpful comment. We have decided to change the title to highlight the fact that our study describes the immediate consequences of chromosome mis-segregation on genome stability.

Minor comments

The schematics of the experimental design are helpful, but are often missing details about the cell cycle synchronization and the amount of time between different steps. Please include these.

We thank the Reviewer for this helpful comment. We followed her/his suggestion and have included the details about the cell cycle synchronization and the amount of time between different steps.

Why is the rate of translocations so high in control aneuploid cells (~30%, Fig. 4g)? Simply inhibiting the SAC for 1 mitosis would not be expected to cause this. Perhaps the cells were analyzed after several days and this is a result of subsequent damage, but the timing with which this experiment was done is not apparent.

We thank the Reviewer for highlighting this. The data shown in Fig. 4g refer to cells harvested after the 2nd mitosis, thus they have been through S and M after chromosome mis-segregation, thus accumulating damage and translocations (that were exacerbated upon DDKi and MiDASi). The experimental procedure for this experiment is outlined in the text (lines 260-266) and in Fig.4a.

The word "length" is misspelled multiple times in Fig. 3a and in Fig. 3c.

Thanks, we fixed this.

Reviewer #4 (Remarks to the Author):

The manuscript by Santaguida and colleagues describes the strategies used by aneuploid cells to protect themselves from excessive replication stress. The authors provide evidence that a combination of DKK-dependent origin firing and MiDAS-driven protection against under-replication suppress aneuploidy-induced chromosome instability. The findings are interesting and relevant to our understanding of aneuploidy in cancer.

We thank the Reviewer for acknowledging that “the findings are interesting and relevant to our understanding of aneuploidy in cancer”.

The MiDAS analysis using individual metaphase chromosomes showed an atypical and unusual pattern of EdU foci. Most foci seem to be individual and located apparently at or very close to the ends of the chromosomes. It would be informative to identify whether the major loci affected by the aneuploidy are telomeric or located at defined locations on specific chromosomes. This could be achieved by combining EdU analysis with either immunofluorescence for TRF2 or PNA FISH for telomeric repeat DNA.

We thank the Reviewer for suggesting this experiment. We followed her/his suggestion and have looked at the location of EdU foci in mitosis. By doing so, we did not spot a tendency of MiDAS to occur at defined chromosomal locations such as telomeres, similarly to aphidicolin-induced MiDAS. These results are shown in Extended Data Fig. 4h,i and mentioned in the text (lines 235-236).

REVIEWERS' COMMENTS

Reviewer #1 (Remarks to the Author):

The authors have addressed the vast majority of the concerns and I support the publication of the paper.

Reviewer #2 (Remarks to the Author):

In the revised version of the manuscript, I consider the authors made substantial effort to address all the major concerns raised upon the first round of revision. In particular:

- the authors have now included a more detailed validation of the drug treatments used regarding cell cycle stage synchronization and excluding any artifacts of the drug treatments in the DNA damage parameters measured.
- the authors have included in Fig.1d the quantification of W-CIN and S-CIN that revealed that numerical aneuploidies accounted for most of the measured aneuploidy events in the 1st and 2nd mitosis. Although, and perhaps worthwhile to mention, high levels of S-CIN are shown later on in Fig. 4e-g for G1 cells after 2nd mitosis. This is likely due to the fact that FISH karyotyping sensitivity is different in metaphase spreads vs G1 spreads
- the authors showed that senescence levels increase upon the 2nd mitosis (Ext. Data Fig. 7)
- the authors demonstrated that aneuploid cells are more sensitive to DDK inhibition; yet note that DDK inhibition impact is modest both in terms of cell viability and DNA damage markers (Fig. 2). Only when both DDK and MiDAS are inhibited, the impact becomes striking
- the authors included a live-cell image analysis with the RNF168-miRFP marker to demonstrate that asymmetric DNA damage distribution correlates with lower proliferative capacity and acknowledged that other aneuploidy-driven events might be involved
- the authors included the comparative analysis between aneuploid cycling and untreated controls in the transcriptome analysis
- the authors acknowledged that this study is on the immediate consequences of aneuploidy which can be different in long-term adapted aneuploid cancer cells.

This said, it is my opinion that the quality of the manuscript has been significantly improved and that a large number of additional experiments have been included that strengthened the conclusions. I also appreciated that the authors carefully addressed all the other relevant points raised by the other Reviewers. The schematic illustrations and graphs have been improved for clarity. Gene modulations have been added to corroborate the drug treatment results. I am therefore supportive of the publication of the manuscript and congratulate the authors for their effort with the revisions.

Reviewer #3 (Remarks to the Author):

The revised manuscript from Garribba et al is substantially improved. This manuscript corroborates previous findings that aneuploid cells have increased replication stress, increased length of S phase, reduced DNA replication fork rate, increased fork stalling, and increased reliance on DDK. The change in title is appropriate. Several additional changes are warranted.

The summary emphasizes that aneuploidy can trigger CIN. However, the manuscript really focuses on 1) structural CIN as opposed to whole chromosome CIN and 2) how cells that continue to cycle limit structural CIN. Thus, there is only a modest increase in structural CIN observed between the 1st and 2nd mitosis after Mps1 treatment. S-CIN is increased by a few percent but W-CIN is reduced so that overall CIN remains constant (Fig. 1d). The abstract should be revised to indicate that they show that aneuploidy can trigger structural CIN and to emphasize that cells counteract this to keep CIN levels low.

In the sentence that begins on line 88, I believe the authors mean to say "Finally, we speculate that DDK-mediated origin firing and MiDAS are crucial for limiting DNA damage and interfering

with those pathways might provide novel therapeutic interventions in cancer therapy.”

In the last line of the Introduction, they authors suggest that their work could help identify patients who could benefit from a DDK inhibitor in clinical trials based on proliferative capacity. It is not clear how that would work. A little elaboration on this point would be beneficial.

I can't tell what was done from the experimental design schematic in Fig. 1a. After the 24h thymidine block it indicates that the cells were treated with Mps1I or DMSO for 24h. But to do M-FISH the cells must have been arrested in mitosis with something that isn't indicated. Between the 1st mitosis and 2nd mitosis in the schematic it says "Wash-out (24h)", which is consistent with whatever that was being washed out, but it doesn't indicate what it was or how long the cells were treated with it. I can't find this info in the text or legend either.

Since the scale in Fig. 1c and ED1i starts at 60 and goes to 100, it is unclear whether the 60% of cells not shown have 1-5 or 0 abnormal events.

Are the control and aneuploid images of RPA in Fig. 1j reversed? From the color images, there appear to be more RPA foci in the control, though the quantitation and text conclude the opposite. (The black and white images look very similar between control and aneuploid.)

There are several conversion errors in Figure 3A, 4A, 6A, 6D, 7 in which the intended letter/symbol is replaced with a box.

Reviewer #4 (Remarks to the Author):

The authors have revised the manuscript to my satisfaction.

REVIEWERS' COMMENTS

Reviewer #1 (Remarks to the Author):

The authors have addressed the vast majority of the concerns and I support the publication of the paper.

We are grateful to the reviewer for all her/his comments that improved the manuscript and for supporting the publication of the paper.

Reviewer #2 (Remarks to the Author):

In the revised version of the manuscript, I consider the authors made substantial effort to address all the major concerns raised upon the first round of revision. In particular:

- the authors have now included a more detailed validation of the drug treatments used regarding cell cycle stage synchronization and excluding any artifacts of the drug treatments in the DNA damage parameters measured.
- the authors have included in Fig.1d the quantification of W-CIN and S-CIN that revealed that numerical aneuploidies accounted for most of the measured aneuploidy events in the 1st and 2nd mitosis. Although, and perhaps worthwhile to mention, high levels of S-CIN are shown later on in Fig. 4e-g for G1 cells after 2nd mitosis. This is likely due to the fact that FISH karyotyping sensitivity is different in metaphase spreads vs G1 spreads
- the authors showed that senescence levels increase upon the 2nd mitosis (Ext. Data Fig. 7)
- the authors demonstrated that aneuploid cells are more sensitive to DDK inhibition; yet note that DDK inhibition impact is modest both in terms of cell viability and DNA damage markers (Fig. 2). Only when both DDK and MiDAS are inhibited, the impact becomes striking
- the authors included a live-cell image analysis with the RNF168-miRFP marker to demonstrate that asymmetric DNA damage distribution correlates with lower proliferative capacity and acknowledged that other aneuploidy-driven events might be involved
- the authors included the comparative analysis between aneuploid cycling and untreated controls in the transcriptome analysis
- the authors acknowledged that this study is on the immediate consequences of aneuploidy which can be different in long-term adapted aneuploid cancer cells.

This said, it is my opinion that the quality of the manuscript has been significantly improved and that a large number of additional experiments have been included that strengthened the conclusions. I also appreciated that the authors carefully addressed all the other relevant points raised by the other Reviewers. The schematic illustrations and graphs have been improved for clarity. Gene modulations have been added to corroborate the drug treatment results. I am therefore supportive of the publication of the manuscript and congratulate the authors for their effort with the revisions.

We are thankful to Reviewer #2 for all her/his insightful suggestions, for recognizing the efforts that have been made to improve the manuscript and for being supportive of the publication of the manuscript.

Reviewer #3 (Remarks to the Author):

The revised manuscript from Garriba et al is substantially improved. This manuscript corroborates previous findings that aneuploid cells have increased replication stress, increased length of S phase, reduced DNA replication fork rate, increased fork stalling, and increased reliance on DDK. The change in title is appropriate. Several additional changes are warranted.

We thank Reviewer #3 for all her/his comments during the revision that have improved our manuscript.

The summary emphasizes that aneuploidy can trigger CIN. However, the manuscript really focuses on 1) structural CIN as opposed to whole chromosome CIN and 2) how cells that continue to cycle limit structural CIN. Thus, there is only a modest increase in structural CIN observed between the 1st and 2nd mitosis after Mps1 treatment. S-CIN is increased by a few percent but W-CIN is reduced so that overall CIN remains constant (Fig. 1d). The abstract should be revised to indicate that they show that aneuploidy can trigger structural CIN and to emphasize that cells counteract this to keep CIN levels low.

We thank the Reviewer for this suggestion. We agree and have elaborated on this in the abstract (line 33) and in the introduction (lines 81 and 82).

In the sentence that begins on line 88, I believe the authors mean to say "Finally, we speculate that DDK-mediated origin firing and MiDAS are crucial for limiting DNA damage and interfering with those pathways might provide novel therapeutic interventions in cancer therapy."

We thank the Reviewer for bringing this up. We agree with this comment and have edited the text accordingly (lines 90-92).

In the last line of the Introduction, they authors suggest that their work could help identify patients who could benefit from a DDK inhibitor in clinical trials based on proliferative capacity. It is not clear how that would work. A little elaboration on this point would be beneficial.

We made this change in the introduction, as suggested by the Reviewer (lines 94-96).

I can't tell what was done from the experimental design schematic in Fig. 1a. After the 24h thymidine block it indicates that the cells were treated with Mps11 or DMSO for 24h. But to do M-FISH the cells must have been arrested in mitosis with something that isn't indicated. Between the 1st mitosis and 2nd mitosis in the schematic it says "Wash-out (24h)", which is consistent with whatever that was being washed out, but it doesn't indicate what it was or how long the cells were treated with it. I can't find this info in the text or legend either.

This info is present in the Methods section (lines 529-536). We are making this clear by indicating in the legend of Figure 1 that more details can be found in the Methods (line 1104).

Since the scale in Fig. 1c and ED1i starts at 60 and goes to 100, it is unclear whether the 60% of cells not shown have 1-5 or 0 abnormal events.

We thank the Reviewer for raising this important point. 60% of cells not shown have 1-5 abnormal events. We are clearly indicating this in Figure legend by stating: "Y axis shown from 60% to 100% for clarity (from 0 to 60% - and above - cells have 1-5 abnormal events, as indicated in key legend)". This is now in lines 1106-1107.

Are the control and aneuploid images of RPA in Fig. 1j reversed? From the color images, there appear to be more RPA foci in the control, though the quantitation and text conclude the opposite. (The black and white images look very similar between control and aneuploid.)

We checked the images and they were not reversed.

There are several conversion errors in Figure 3A, 4A, 6A, 6D, 7 in which the intended letter/symbol is replaced with a box.

We fixed this.

Reviewer #4 (Remarks to the Author):

The authors have revised the manuscript to my satisfaction.

We thank Reviewer #4 for her/his comments during the revision of our manuscript.